# A signal-like role for floral humidity in a nocturnal pollination system

**Ajinkya Dahake** [1] ✉, **Piyush Jain**[2], **Caleb C. Vogt** [1], **William Kandalaft** [1], **Abraham D. Stroock**[3] **& Robert A. Raguso**[1]

Previous studies have considered floral humidity to be an inadvertent consequence of nectar evaporation, which could be exploited as a cue by nectar-seeking pollinators. By contrast, our interdisciplinary study of a night-blooming flower, *Datura wrightii*, and its hawkmoth pollinator, *Manduca sexta*, reveals that floral relative humidity acts as a mutually beneficial signal in this system. The distinction between cue- and signal-based functions is illustrated by three experimental findings. First, floral humidity gradients in *Datura* are nearly ten-fold greater than those reported for other species, and result from active (stomatal conductance) rather than passive (nectar evaporation) processes. These humidity gradients are sustained in the face of wind and are reconstituted within seconds of moth visitation, implying substantial physiological costs to these desert plants. Second, the water balance costs in *Datura* are compensated through increased visitation by *Manduca* moths, with concomitant increases in pollen export. We show that moths are innately attracted to humid flowers, even when floral humidity and nectar rewards are experimentally decoupled. Moreover, moths can track minute changes in humidity via antennal hygrosensory sensilla but fail to do so when these sensilla are experimentally occluded. Third, their preference for humid flowers benefits hawkmoths by reducing the energetic costs of flower handling during nectar foraging. Taken together, these findings suggest that floral humidity may function as a signal mediating the final stages of floral choice by hawkmoths, complementing the attractive functions of visual and olfactory signals beyond the floral threshold in this nocturnal plant-pollinator system.

The spatial scale at which pollinators are attracted by floral traits has important consequences for pollinator foraging efficiency[1], resource partitioning[2], and plant gene flow[3,4]. Although floral scent and color can attract pollinators at a scale of meters[5–9], they cease to be informative once pollinators arrive at a flower's threshold (mm to cm distance), in the absence of additional information, such as contrasting nectar guides[10], scented pollen[11] or nectar[12]. Recently foraged flowers can remain scented, turgid, and pigmented minutes to hours after

nectar or pollen has been removed by an earlier visitor, yet it is commonly observed that pollinators reject some flowers upon inspection, without landing[13,14]. Pollinators likely make such decisions at a short range from the flowers based on more reliable sources of information as they navigate a patch of flowering plants[15,16]. For instance, floral primary metabolism and transpiration produce gradients in carbon dioxide ($CO_2$) concentration and relative humidity (RH) within the headspace of a newly opening flower (mm to cm distance), which more

[1]Department of Neurobiology and Behavior, Cornell University, Ithaca, NY 14853, USA. [2]Sibley School of Mechanical and Aerospace Engineering, Cornell University, Ithaca, NY 14853, USA. [3]Smith School of Chemical and Biomolecular Engineering, Cornell University, Ithaca, NY 14853, USA. ✉e-mail: asd244@cornell.edu

reliably indicate nectar availability before pollinators commit to probing or visiting a flower[17,18].

All animals utilize cues—the sensory information available in their environments—to navigate and survive. When cues are produced inadvertently by the movement or metabolism of other organisms, they can be exploited by eavesdroppers[19]. In the context of communication, when a respiring animal (sender) exhales $CO_2$, it alerts nearby mosquitoes (receivers) to a potential blood meal, with detrimental consequences for the sender of the cue[20]. In contrast, signals mediate communication between senders and receivers that results, on average, in fitness benefits to both parties[19,21]. Despite longstanding debate on signal classification and evolution[22,23], behaviorists distinguish signals from cues using the following criteria: (1) senders provide clear, measurable information, (2) that has evolved for the purpose of communication with receivers, which (3) elicit a distinctive (e.g. state-altering) response from the recipient, resulting in (4) fitness consequences that are favorable, on average, to both parties[19,22,23]. In addition, although exceptions exist, most signals incur significant metabolic, social- or health-related costs that are thought to ensure evolutionary stability against cheating[24]. In plant–pollinator communication, floral signals may evolve as "indices" (form and content are physically connected), as "icons" (the form is similar to the content but can be decoupled), or as "symbols" (form and content are arbitrarily or statistically linked[22]).

Current evidence suggests that floral $CO_2$ is a cue by which pollinators can assess nectar presence and profitability. Night-blooming flowers such as *Datura wrightii* accumulate and release $CO_2$ upon anthesis when nectar is most available, but floral $CO_2$ decays to ambient levels soon after anthesis. Thus, the above-ambient floral $CO_2$ alerts pollinators to the presence of newly opened, profitable flowers[25]. Flowers with above-ambient $CO_2$ are more attractive to hawkmoths (*Manduca sexta*) than those with ambient $CO_2$[26], but in the absence of a sustained difference in profitability, hawkmoth preference for high $CO_2$ diminishes to chance over subsequent floral visits[27]. Hawkmoths (as receivers) can utilize above-ambient floral $CO_2$ as an ephemeral profitability cue for freshly opened flowers, due to its correlation with unexploited floral rewards. It remains unclear whether plants benefit by furnishing floral $CO_2$ as a cue, as they might derive greater fitness benefits by withholding nectar profitability information[28].

Unlike $CO_2$, floral humidity appears to indicate nectar presence to foraging pollinators as a direct physical consequence of nectar evaporation, rather than as a correlated aspect of anthesis. Unlike $CO_2$, floral humidity may therefore alert pollinators to flowers that refill nectar after anthesis. The evening primrose flower (*Oenothera cespitosa)* presents 4–6% above-ambient RH in its headspace, which decays to ambient levels within 30 min after anthesis[29]. Floral manipulations revealed that nectar evaporation accounts for half of the floral humidity in *O. cespitosa*, strongly suggesting a function as a cue. A recent survey[30] of floral humidity from 42 plant species in a common garden reported a range of 0.05–3.7% above-ambient RH (henceforth ΔRH) in floral headspace, including species that lack floral nectar[31]. In the laboratory, *Hyles lineata*, a common pollinator of *O. cespitosa*, prefers probing non-rewarding artificial flowers with above-ambient RH over flowers with ambient RH, despite the absence of sugar rewards[29]. Finally, the generalist bumblebee, *Bombus terrestris*, has been shown to discriminate ΔRH on artificial flowers when paired with sugar rewards in a lab setting[32]. Together, these findings indicate that floral humidity influences pollinator foraging decisions, but also suggest that our view of floral RH as an unavoidable consequence of nectar evaporation may be overly simplistic.

Despite growing acceptance that floral humidity can serve as an additional pollinator attractant, there are several gaps in our understanding of the proximate mechanisms governing RH production by flowers and perception by pollinators, the first and third criteria for signal definition, respectively[23]. Some of these gaps include the physiological sources of floral RH, the efficacy (physical robustness) of floral RH gradients in the face of environmental noise[33], and the mechanisms of pollinator perceptual and behavioral responses to realistic RH gradients in space and time. In contrast to the rapid dissipation (~30 min) of floral RH from the narrow nectar tube and open petals of *O. cespitosa*[29], we hypothesized that larger, trumpet-shaped corollas (e.g., of *Datura* flowers) might sustain humidity gradients beyond anthesis (also see ref. 30). Furthermore, if above-ambient floral RH persists after nectar has been extracted by an earlier visitor, the disconnect between floral humidity and nectar status may present conflicting information to subsequent floral visitors. If floral humidity and nectar are physiologically decoupled, this may expand the possible roles of floral RH from a profitability cue for pollinators to an icon signal, given the form-content relationship between plant water balance, nectar secretion, and the provision of humid air as information. Signals are thought to evolve from cues when selection favors the increased size or intensity of the trait with an attendant increase in receiver response[19]. In the case of icons, increased signal magnitude often incurs a sizable cost to the sender, which can be offset by the fitness benefits of a concomitant increase in responsiveness by the receiver[19].

We evaluate the role of floral humidity in the well-studied mutualistic relationship between *Datura wrightii*, a night-blooming plant with large, trumpet-shaped flowers, and *Manduca sexta*, a nocturnal hawkmoth that is the primary pollinator of *Datura*[34,35]. We show that *Datura* flowers present unusually high floral humidity (>30% ΔRH), exceeding levels previously reported for angiosperm flowers[29,30] by a factor of 10. We find that *Datura* floral humidity is not a passive consequence of nectar evaporation but, instead, is a persistent floral trait that results from gas exchange through floral stomates. Neurophysiological responses from *Manduca* antennal hygrosensing neurons confirm that moths perceive minute differences in floral humidity, and experimental occlusion of the hygrosensing sensillum abolishes their innate behavioral preference for humid flowers. We experimentally decouple floral humidity from nectar presence in artificial flowers, measuring moth behavioral responses to three treatments representing alternative functions for floral humidity: an uninformative trait, an informative trait positively associated with nectar, or an informative trait negatively associated with nectar. Our results show that floral humidity benefits *Datura* plants through increased pollinator visitation and benefits *Manduca* moths by reducing energetic expenditure related to flower handling time. Moths' strong innate preference for humid flowers persisted irrespective of the presence or absence of nectar in the artificial flowers. Furthermore, the maintenance of unusually high ΔRH in a desert/grassland plant facing severe challenges in water balance suggests that floral RH in *Datura* is a costly sexual signal, evocative of an overstated animal courtship display[36]. In summary, using a neuroethological approach combined with floral physiology, animal behavior, and sensory ecology, we find that the functional role of floral humidity in this nocturnal pollination system is complementary to other attractive signals like floral scent and color, but is more spatially relevant at the threshold of the flower. We discuss the range of possible roles floral humidity can play as an informative trait beyond the (limited) current view that it is a cue for the presence of nectar.

## Results
### *Datura* flowers exhibit a large and consistent humidity gradient
At what spatial scale might floral RH be a relevant stimulus, and for how long after flowers open? Flowers of *Datura wrightii* exhibit an appreciable vertical humidity gradient (ΔRH) within the floral tube, with the greatest ΔRH observed at the base of the flower tube (= 0 mm) (mean ± SEM, 31.01 ± 1.20 %, $n = 38$), persisting across a broad range of background ambient RH (Fig. 1a). At the opening of the flower, ~70 mm above the corolla base (midpoint of the transect), the ΔRH was

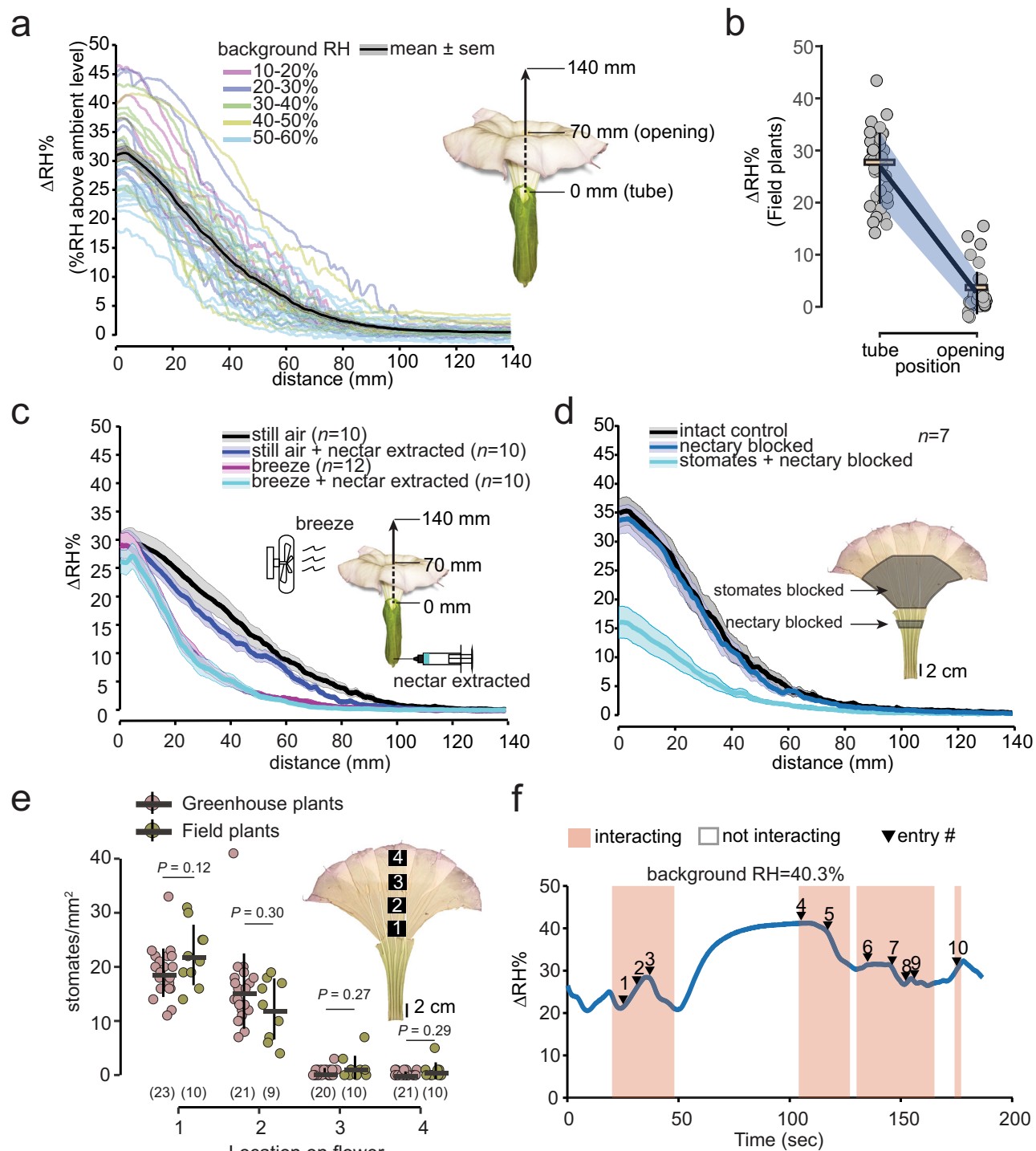

4.08 ± 0.50% ΔRH (Fig. 1a). Horizontal transects taken at the flower opening showed 3.89 ± 0.42% ΔRH at the midpoint, consistent with the ΔRH recorded for the vertical transect at the same location (Supplementary Fig. 1a). At 140 mm above the flower tube, floral humidity was only marginally higher (0.46 ± 0.16% ΔRH) than the background. Within the floral tube, the highest ΔRH (38.82 ± 2.90%; $n = 4$) was recorded when ambient RH was 10–20%, while the lowest ΔRH (24.16 ± 0.76%; $n = 14$) was observed when ambient RH was 50–60%. We compared the floral humidity curves across background RH levels using two model parameters: the decay rate ($\alpha$) of the curve and the intercept $y0$ (see Methods). Multiple comparisons suggested that the intercept ($y0$) differs across the range of background humidity but that

the decay rate ($\alpha$) does not (Supplementary Fig. 2 and Supplementary Tables 1, 2). Specifically, floral humidity measured when background RH was 50–60% was much lower than when background RH was 10–20% ($t = 3.92$, $P = 0.001$), 20–30% ($t = 3.50$, $P = 0.004$), and 30–40% ($t = 4.32$, $P = 0.0001$). No other differences were found for floral humidity at any other levels of background RH.

To confirm that the unusually high floral humidity in *Datura* is not an artifact of greenhouse conditions, we sampled from wild *D. wrightii* plants growing in their natural habitat at multiple locations near Tucson, Pima Co., Arizona, USA (Aug. 2019; see Methods for site details). Even under field conditions and at backgrounds of 20–40% ambient RH, we recorded 26.58 ± 6.71% (mean ± SD) ΔRH in the flower tube and

**Fig. 1 | The structure, efficacy, and source of floral humidity as a potentially informative trait for foraging pollinators. a** Summary of the multiple transects of floral humidity measured from the base of the flower tube (0 mm) to outside the flower opening (140 mm) in a range of background humidity (10–60% RH). See the inset illustration of the vertical transect. Individual flower transects are color-coded by the background humidity they were measured at. The solid black line shows the mean ± SEM (gray shading) of *n* = 38 individual flowers. (Some transects are pooled from several experiments that are shown below). **b** Floral humidity of *n*=37 naturally growing *Datura* flowers from Tucson, Arizona, USA, shows that it is not an artifact of greenhouse conditions. Floral RH is measured at two positions: flower opening (70 mm) and at the tube base (0 mm). Gray dots show individual flowers and line plots show the mean and SD (blue-shaded area). **c** Comparing the effect of breeze (-0.4 m/s) and/or nectar extraction on the floral humidity. Nectar extraction does not impact the floral RH gradient; however, breeze affects the decay, but not the intercept of the humidity curve. The inset figure illustrates the method for the different treatments. Treatments are color-coded, showing mean (bold lines) ± SEM

(shaded area) with sample sizes in parentheses. **d** Effect of nectary and stomatal blockage, as illustrated (inset), on the floral humidity curves. Nectary blockage does not impact the floral RH gradient, but stomatal blockage halves the RH gradient. The mean (solid lines) ± SEM (shaded) are color-coded by treatment (*n* = 7 for each treatment). **e** Stomatal counts across 4 locations on the inner surface of the flower (see inset diagram) on greenhouse-grown plants (magenta) and field plants from Tucson, Arizona, USA (olive green). Dot plots show counts from individual flowers, black line plots show the mean (horizontal) ± SD (vertical) with sample sizes in parentheses. A two-tailed Mann–Whitney test is performed on the stomatal counts between the same locations on the flower with *P* values shown on top. Data suggest that the stomatal distribution is not specific to where the plants grow. **f** Exemplary trace of floral humidity (solid blue line) measured continuously in the flower tube while moths interact with the flowers (orange shading) or enter the flower tube multiple times while probing (black triangles with the entry number). Floral humidity reconstitutes within seconds when the moth is not interacting with the flower. Source data are provided as a Source Data file.

2.64 ± 4.02% ΔRH at the flower opening (Fig. 1b and Supplementary Table 3).

## Floral ΔRH persists despite ambient disturbance or nectar depletion

How robust are floral humidity gradients to wind and other disturbances? We performed a series of floral manipulations to test the efficacy of floral humidity under natural settings. We sampled floral humidity in still air as a control and subsequently added a gentle breeze to evaluate its effect on floral humidity (Fig. 1c). Compared with flowers in still air, the experimental breeze attenuated the vertical gradient of floral humidity ($\alpha$:*t* = −6.60, *P* < 0.0001; Supplementary Table 4), but had no impact within the floral tube ($y0$: *t* = 0.72, *P* = 0.88; Supplementary Table 5). We then extracted floral nectar to simulate moths probing and emptying the flowers. We found no evidence that nectar depletion influences floral humidity when comparing flowers sampled in still air with or without nectar (Fig. 1c). The decay rate ($\alpha$) and the intercept ($y0$) of the floral humidity gradient were statistically indistinguishable between the still air and still air + nectar-extracted treatments ($\alpha$: *t* = −1.42, *P* = 0.48; $y0$: *t* = 0.53, *P* = 0.95; Supplementary Tables 4, 5). Finally, we subjected flowers to both breeze and nectar depletion but saw no difference in the decay rate or the intercept in comparison to the flowers in the breeze with nectar present ($\alpha$: *t* = 0.08, *P* = 0.99; $y0$: *t* = 0.53, *P* = 0.95; Supplementary Tables 4, 5). Overall, these results indicate that the presence of floral nectar is not sufficient to generate the observed floral RH gradients.

We hypothesized that floral humidity in *Datura* may result from the accumulation of saturated air in the floral headspace through development from the bud stage and that dissipation of the humidity is prevented by the conical architecture of the flower. Accordingly, we predicted that a moth's visit to the flower should deplete floral humidity due to the rapid (~25 Hz)[37] wing fanning of a hovering moth. To test this prediction, we allowed moths to forage on newly opened *Datura* flowers while simultaneously recording the humidity in the floral tube (Supplementary Movie 1). Floral humidity never decayed to ambient levels, even as moths hovered at the flower opening (interact) or entered flowers while probing (entry#). Remarkably, floral humidity was reconstituted to previous levels within 30 s of moth departure (Fig. 1f).

## Floral transpiration accounts for the majority of floral humidity

If not nectar, what is the primary source of humidity in *Datura* flowers? We conducted a separate experiment to evaluate the relative contributions of nectar evaporation and floral transpiration to floral humidity in *Datura*. We first blocked the nectary with petroleum jelly and compared the floral humidity transect with unmanipulated flowers. As expected, nectary blockage did not impact floral humidity

curves (Fig. 1d). The decay rate and the intercept of the floral humidity were identical for the control and nectary blocked flowers, as noted from the fitted model predictions ($\alpha$:*t* = −0.10, *P* = 0.99; $y0$: *t* = 0.26, *P* = 0.96; Supplementary Fig. 4 and Supplementary Tables 6, 7). However, when both the nectary and the inner corolla surface were blocked with petroleum jelly, the magnitude of ΔRH was halved (Fig. 1d). The humidity curves of the flowers with their nectary and stomates blocked showed a significantly smaller intercept than the other two treatments ($y0$: t = 5.60, *P* < 0.0001; and $y0$: t = 5.34, *P* < 0.0001; Supplementary Table 7), but the $\alpha$ did not differ significantly ($\alpha$: t = −0.52, *P* = 0.86; & $\alpha$: t = −0.41, *P* = 0.90; Supplementary Table 6). To isolate the contribution of only standing nectar pools to floral humidity, we added 200 µl of *Datura* nectar or water to an artificial flower and measured humidity transects. Peak floral humidity through nectar or water evaporation was only 2.44 ± 0.33% ΔRH (mean ± SEM) for nectar and only 3.16 ± 0.61% ΔRH for water (Supplementary Fig. 1c, d) at the base of the flower tube. These results, combined with the nectar removal experiments above, demonstrate that transpiration is likely the major source of floral humidity in *Datura wrightii*.

## Floral stomatal distribution aligns with the humidity gradient

Do flowers contain stomates on the corolla to facilitate water vapor emission, as leaves do? Floral peels across four locations from the tube base to the inner (adaxial) corolla limb (Fig. 1e inset) indicated that stomates were found within the corolla (Supplementary Fig. 5) at high density near the tube base but were scarce to absent towards the distal limb (Fig. 1e). This pattern was consistent between flowers of greenhouse-grown and wild plants. Stomates were observed across all four zones sampled on the outer (abaxial) corolla surface, and mean stomatal density was greater for wild plants than for greenhouse-grown plants (Supplementary Fig. 1b). These data support the hypothesis that physiological gas exchange, rather than nectar evaporation, is responsible for the steep floral humidity gradients we have measured in the laboratory and the field.

## Anatomy of the hygrosensing sensillum

How do moths sense floral humidity gradients? Previous anatomical surveys identified at least two classes of aporous putative hygrothermosensory sensilla on both male and female antennae of *Manduca*[38,39]. The coeloconic type B is a small (2 µm) peg-in-pit sensillum not easily visualized with microscopy. In contrast, the styliform complex is a large (30–40 µm), flexible-peg-type sensillum on the leading edge of each antennal annulus and is easily distinguishable from other sensory pegs (Fig. 2a). The styliform complex is the largest sensillum on female *Manduca* antennae, whereas it is surrounded by many large (pheromone-detecting) trichoid sensilla on male *Manduca* antennae (Fig. 2b, also see ref. 40). The tip of each

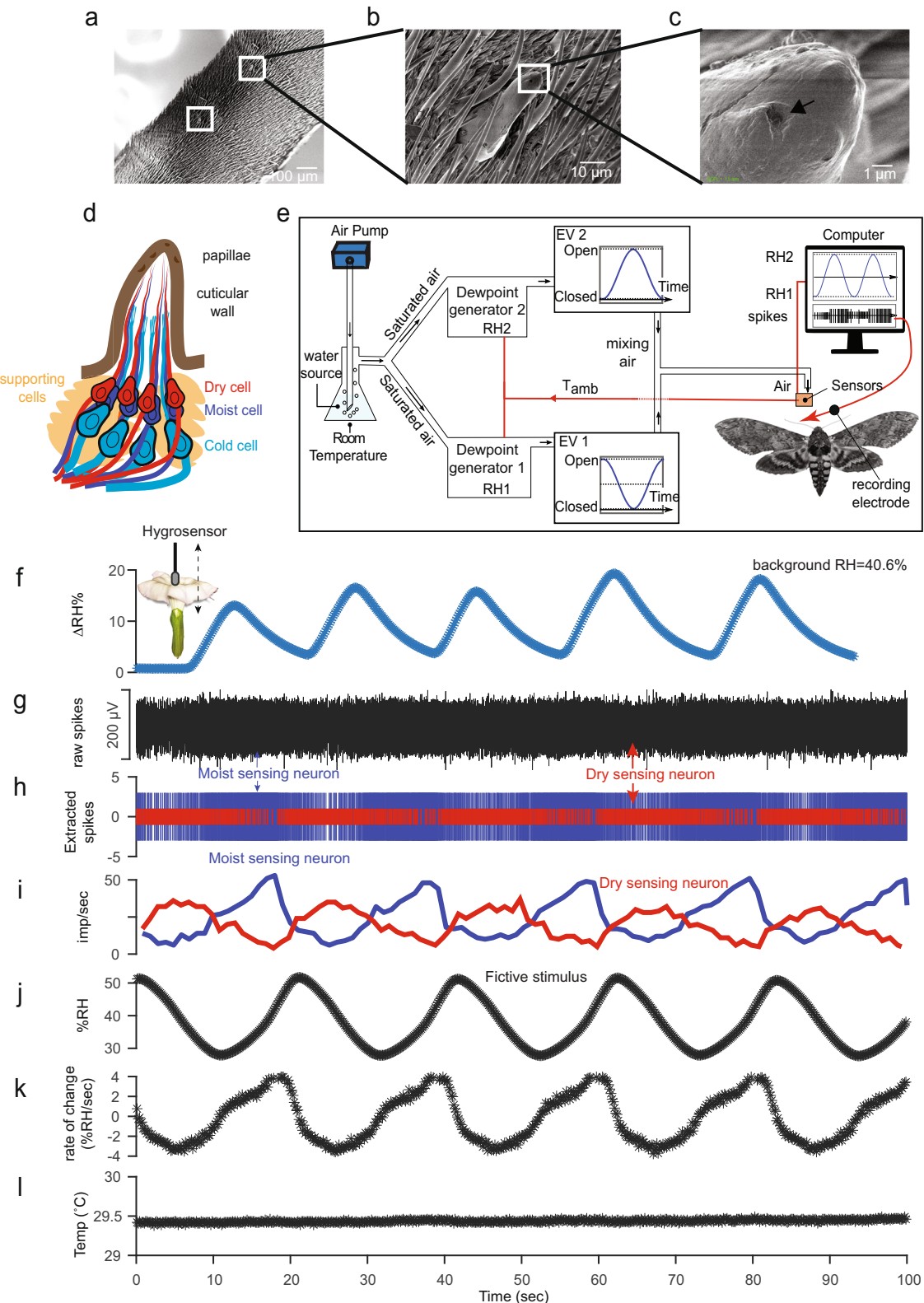

sensillum houses 3–5 papillae (Fig. 2c) of 2 μm diameter each. Individual papilla house 3 dendrites; 2 cylindrical, and 1 lamellate type, as one unit enclosed within a dendritic sheath, typical of hygro-thermosensory function (see refs. 39, 41). Thus, 9–15 dendrites innervate each putative humidity sensing organ, depending on the number of papillae at the tip of the organ, repeated over ~80 annuli in both antennae (Fig. 2d).

**Generating a floral humidity stimulus as experienced by moths**

Although insects can sense ambient humidity, it is unclear whether the range of floral humidity is sufficient to trigger robust responses by the hygrosensory neurons of pollinators. Simulating how moths probe *Datura* flowers, the hygrosensing probe was dipped in and out of the corolla to generate the range of humidity changes experienced by the hygrosensors on the moth antenna. Figure 2f shows the sinusoidal

**Fig. 2 | Moth antennal hygrosensory neurons respond to the range of floral humidity presented by *Datura* flowers. a** Zoomed view sequence of electron microscopy images of the styliform complex sensillum on *Manduca sexta* antenna. The scale bar is shown at the bottom right corner of each image. The image shows two segments of a female antenna and white squares show the location of the styliform complex sensilla on the leading edge of the antenna. **b** Zoomed view of the entire styliform sensillum surrounded by trichoid sensilla. **c** Zoomed view of the tip of the styliform sensillum. Black arrow points toward one of the papillae. **d** A representation of the longitudinal section of the styliform sensillum showing the underlying dendrites and cell bodies based on TEM[39,40] and cryosections of the organ. **e** Schematic of the stimulus delivery setup. Water vapor saturated air at room temperature is sent to two different dewpoint generators, outputting air with fixed relative humidity, *RH1* and *RH2*, corresponding to the ambient temperature, $T_{amb}$ measured adjacent to the moth. Two electric valves (EV) operated by motors at the outlet of the dewpoint generators regulate the mass flow rate in an antiphase synchronized manner (as shown), which is sent to a thermostatic mixing valve to deliver air with a sinusoidally varying humidity airstream, like the fictive stimulus in

(**j**). Temperature and RH were measured using sensors placed adjacent to the moth antennae. A tungsten electrode was inserted at the sensillum base for electrophysiology. Electrical wirings are denoted in red, and black arrows denote the direction of airflow (see Methods for details). **f** A humidity stimulus generated by dipping the hygrosensing probe in and out of a *Datura* flower (see inset illustration) to mimic the humidity experience of moths probing and entering *Datura* flowers. The blue line shows %ΔRH at 40.6% ambient RH. **g** Exemplary single sensillum recording of the styliform sensillum (a-d) showing simultaneously recorded extracellular activity of moist and dry neurons (arrows) within a single sensillum. **h** An overlaid raster plot of the spikes sorted from the raw trace in (**g**) showing the activity of the moist sensing neuron (blue) and the dry sensing neuron (red). **i** Moving average of the impulse frequency of the moist (blue) and dry (red) sensing neurons. **j** A fictive stimulus of floral RH matching the experience of moths probing *Datura* (shown in f) with the amplitude ranging from 30 to 50% RH and a period of approximately 30 s. **k** Continuous rate of change in RH across the recording period. **l** A constant temperature (°C) is maintained across the recording period. Source data are provided as a Source Data file.

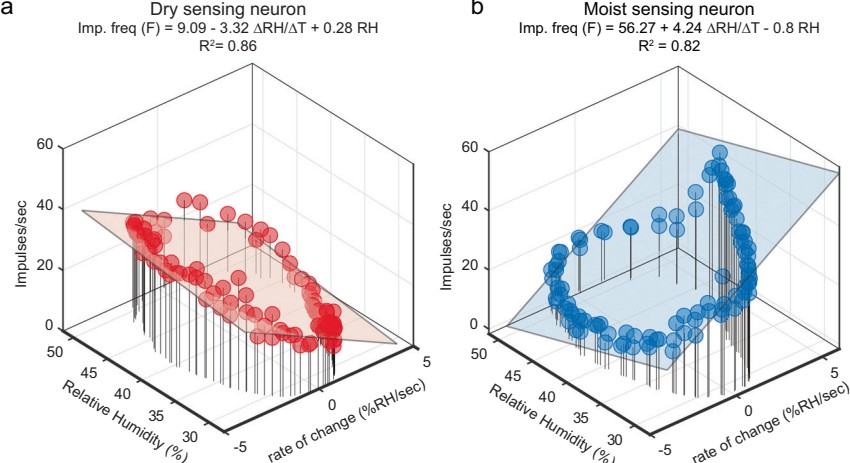

**Fig. 3 | Response properties of the hygrosensory neurons are tuned to the rate of change in humidity.** 3D surface curve fitted scatterplots of the impulse frequency of the dry (**a**) and moist (**b**) neuron plotted against the rate of change in RH and the instantaneous RH. The fitted equation for the polynomial linear regression is shown at the top of each panel, along with their goodness-of-fit measure ($R^2$). Source data are provided as a Source Data file.

change in humidity as measured by the sensor entering and departing as a moth does while approaching and probing a *Datura* flower. At 40% background RH, the sensor measured a rapid increase of 15% ΔRH, which dropped precipitously to ambient levels when removed from the flower.

## Single-sensillum electrophysiology

How can we design a fictive humidity stimulus for moths? In neuroethology, a fictive stimulus captures the essence of a sensory stimulus experienced by an animal in its natural environment and presents it in a controlled laboratory experiment[42]. We fashioned an experimental stimulus that temporally matches how moths might perceive RH as they enter and depart *Datura* flowers (Fig. 2f). The custom-built stimulus delivery system (Fig. 2e) generated a sinewave of RH whose amplitude and frequency could be altered to simulate the humidity change experienced by the hygrosensors on the moth antennae (Fig. 2j). Out of 39 electrophysiological recording events, 28 yielded responses to our stimulus from at least one type of sensory neuron, characterized as "moist", "cold", or "dry". The underlying moist and dry sensing neurons (cold sensing neurons not evaluated here) responded robustly and predictably (Fig. 2g) throughout our sinewave RH stimulus (Fig. 2j). This setup allowed us to maintain a stable temperature of the stimulus air while varying RH (Fig. 2l). The moist and dry neurons were distinguishable based on their amplitudes in most of the recordings (Fig. 2h). The firing frequency of the moist

neuron increased in proportion to the stimulus RH and remained well-correlated with the shape and phase of the rate of change of RH (Fig. 2k). The dry neuron increased firing frequency as the stimulus RH decreased and was -180° out of phase with the firing frequency of the moist neuron (Fig. 2i). Thus, the moist and dry sensing neurons showed the stereotypical antagonistic activity of the hygrosensory neurons previously demonstrated for other arthropods[43–46].

The few studied cases of insect hygrosensory neurons suggest that the response properties of the neurons are a function of both instantaneous humidity and the rate of change in humidity. Typically, the impulse frequencies of the moist sensing neuron increase in proportion to the instantaneous RH and rate of change in RH, whereas the dry sensing neurons respond antagonistically[46,47]. To evaluate the response properties of *Manduca* hygrosensory neurons to these parameters, we fitted the data points with a polynomial linear regression of the form $F = a + b\, \Delta RH/\Delta T + c\, RH$, where $F$ is the impulse frequency of the neuron, $a$ is the height of the regression plane, $b$ is the slope for the rate of change in RH, and $c$ is the slope for the instantaneous RH. For both moist and dry sensing neurons, the slope for the rate of change in RH $b$, was larger than the slope for instantaneous RH, suggesting a higher sensitivity to the rate of change ±4% RH/sec, Figs. 2k and 3a, b) compared to instantaneous RH (30–50%). For a rate of change of +1% RH/sec, this amounts to a decrease of −3.32 imp/sec for the dry sensing neuron, and, correspondingly an increase of +4.24 imp/sec for the moist sensing neuron. For an increase in instantaneous

RH by 1%, this amounts to a sensitivity of +0.28 imp/s for the dry sensing neuron, and −0.80 imp/sec for the moist sensing neuron. Calculations show that an increase of 1 imp/sec in the dry sensing neuron is elicited either by an increase of 3.56% instantaneous RH if the rate of change is constant, or by a rate of change of only −0.30%RH/sec. Similarly, for the moist sensing neuron, an increase of 1 imp/sec is reflected either by −1.24% instantaneous RH, if the rate of humidity change is constant, or by increasing the rate of change by only 0.23% RH/sec. Therefore, both hygrosensing neurons of *Manduca* are more influenced by the rate of humidity change of 1% RH/sec than by a 1% increase in instantaneous RH. This finding is opposite to previous reports on other insects where the slope parameter $c$ was found positive for instantaneous RH for the moist sensing neuron and negative for the dry sensing neuron[46–48]. We conclude that *Manduca's* styliform sensilla house hygrosensory neurons that are sensitive to fluctuations in humidity which moths likely experience as they hover at and enter floral headspace at the scale of flower patches (cm to m)[49] or traverse different habitats at the scale of a landscape (m to km).

## Flower-naïve hawkmoth preference for unrewarded artificial flowers

Do moths show an innate preference for humid flowers? We presented flower-naïve adult *Manduca* with a choice between empty flowers with ambient humidity (henceforth, "ambient flowers") vs. above-ambient humidity (henceforth, "humid flowers") (Fig. 4a). The humid flowers presented a range of ΔRH matching the humidity measured from *Datura* flowers (Fig. 4b and Supplementary Fig. 2). We used video tracking software to measure two response variables: probing duration and the number of floral entries made per visit. For the probing duration, we measured how long the proboscis tip (labeled) was present within a circumscribed perimeter of the flower (region of interest shown in Fig. 4c). Likewise, for the number of floral entries, we tracked how often the moth's eye (labeled) crossed the flower rim (region of interest shown in Fig. 4d). Because moths visited both flowers frequently in all experimental trials, this setup allowed us to measure overnight trends in moth responses toward humid vs. artificial flowers, beyond central tendencies such as their first choice (Supplementary Fig. 9). Neither males nor females showed side bias for probing duration (males: $Z = 0.29$, $P = 0.76$; females: $Z = 1.86$, $P = 0.06$; Fig. 4e) or the number of entries (males: $Z = 0.39$, $P = 0.69$; females: $Z = 1.28$, $P = 0.19$; Fig. 4f) when presented a choice between two identical ambient flowers. Irrespective of sex, moths probed longer on the humid flower than on the ambient flower (males: $Z = 3.85$, $P = 0.0001$; females: $Z = 8.25$, $P < 0.0001$; Fig. 4g) and entered humid flowers more frequently (males: $Z = 5.08$, $P < 0.0001$; females: $Z = 7.22$, $P < 0.0001$; Fig. 4h). To assess if this strong preference for humid flowers is mediated through hygrosensation, we occluded a strip along the leading edge of the moth antennae with UV-hardened glue to block the styliform sensillum from contact with ambient air. Hygrosensor-blocked moths approached flowers less frequently, as noted from the small number of visits shown in Fig. 4I, j. Nevertheless, the moths that did visit the flowers entered both ambient and humid flowers equally (males: $Z = 1.42$, $P = 0.15$; females: $Z = 0.003$, $P = 0.99$; Fig. 4j) and showed no strong preference for probing on either flower (males: $Z = 0.98$, $P = 0.32$; females: $Z = 0.62$, $P = 0.53$; Fig. 4i). Thus, moths with impaired antennal hygrosensation could not differentiate between humid and ambient flowers. To account for the unintended effects of glue on moth antennae, we included a sham treatment where 5–10 annuli of the antennae were coated with the glue (Supplementary Fig. 6b, d). Using racemic linalool as a common floral odorant, we evaluated the electroantennogram response of the whole antennae of the hygrosensor blocked and sham control moths to test whether the antennae are still competent after the glue treatment. EAG responses showed that both treatments retain olfactory sensitivity to linalool (Supplementary Fig. 6a, c). Sham control moths retained a probing

preference for the humid flowers (males: $Z = 4.04$, $P < 0.0001$; females: $Z = 3.43$, $P = 0.0005$; Fig. 4k) and entered humid flowers more frequently, just as unmanipulated moths do (males: $Z = 3.48$, $P = 0.0004$; females: $Z = 3.00$, $P = 0.002$; Fig. 4l). This confirmed that the glue occlusion was local and did not affect normal olfactory responses. In summary, both male and female flower-naïve *Manduca* show innate preferences for humid flowers. Their preference for empty humid flowers persists all night, despite the absence of nectar rewards as positive reinforcement.

## Moth preference for sugar-rewarded artificial flowers

Do moth preferences change in the presence of nectar? We modified the artificial flowers used in our binary-choice assay to selectively include or exclude sugar rewards (see Supplementary Fig. 7b, c for flower design). Because the previous experiment revealed that moths show a strong innate preference for floral humidity (Fig. 4), a null hypothesis ($H_0$) that moths ignore RH when foraging for nectar would be rejected. Instead, we evaluated whether ($H_1$) moths show a fixed preference for floral humidity that cannot be overridden by experience, or ($H_2$) moth preference can be modified in the case of differential profitability (Fig. 5a–c). We presented moths with three experimental treatments, each with a relevant hypothesis, decoupling the presence/absence of sugar rewards with floral humidity. These treatments, their effects on the information content of floral RH and the predicted responses of moths consistent with $H_1$ or $H_2$ are outlined in Table 1. As in the previous experiment, for all treatments, moths visited both flowers in equal proportions throughout the night (Supplementary Fig. 10). For each treatment, we measured the moth's probing duration (sec) and the number of entries in both flowers, which would benefit plants through increased pollen export and therefore male siring success[50]. Likewise, we measured flower handling time (sec) until nectar discovery by moths and used it to calculate net energy gain and expenditure (J) during floral visits as surrogates for moth fitness[51].

In the "no-association treatment", floral humidity is an uninformative trait because both ambient and humid flowers present sugar rewards. Nevertheless, moths showed a strong, persistent preference for the humid flower, supporting the prediction for $H_1$ but not $H_2$ (Table 1 and Fig. 5a). Moths probed longer on the humid flower ($Z = 2.73$, $P = 0.006$; Fig. 5d) and entered humid flowers more frequently ($t = 2.69$, $P = 0.008$; Fig. 5e) resulting in a potential fitness advantage to plants with humid flowers.

In the "positive association" treatment, floral humidity is an informative trait for a positive association between humidity and nectar, because humid flowers have sugar rewards and ambient flowers are empty. Consistent with both $H_1$ and $H_2$, we predicted the humid flower would receive more visits by moths because their preference for humidity and the presence of a reward are in alignment (Table 1 and Fig. 5b). As in the previous treatment, moths probed and entered the humid, rewarding flower significantly more frequently than the empty ambient flower (probing duration: $t = 5.98$, $P < 0.0001$, Fig. 5f; entries: $t = 4.6$, $P < 0.0001$, Fig. 5g).

Finally, in the "negative association" treatment, floral humidity is an informative trait for moths, but for the absence, rather than the presence, of nectar, because a sugar reward was present in the ambient flower, but the humid flower was empty. For $H_1$, we predicted that moths will continue to show a strong preference for the humid flower, whereas, for $H_2$, we predicted that flowers with ambient humidity would experience (and benefit from) increased visitation by moths because, over time, insects can learn trait combinations with rewards, as shown for bumblebees associating dry flowers with nectar presence[52] (Table 1 and Fig. 5c). Contrary to the predictions for both hypotheses, moths did not show a strong preference for either flower (probing duration: $Z = 0.75$, $P = 0.45$, Fig. 5h; entries: $Z = 0.14$, $P = 0.88$, Fig. 5i). This result suggests that despite the payoff disparity, moths did

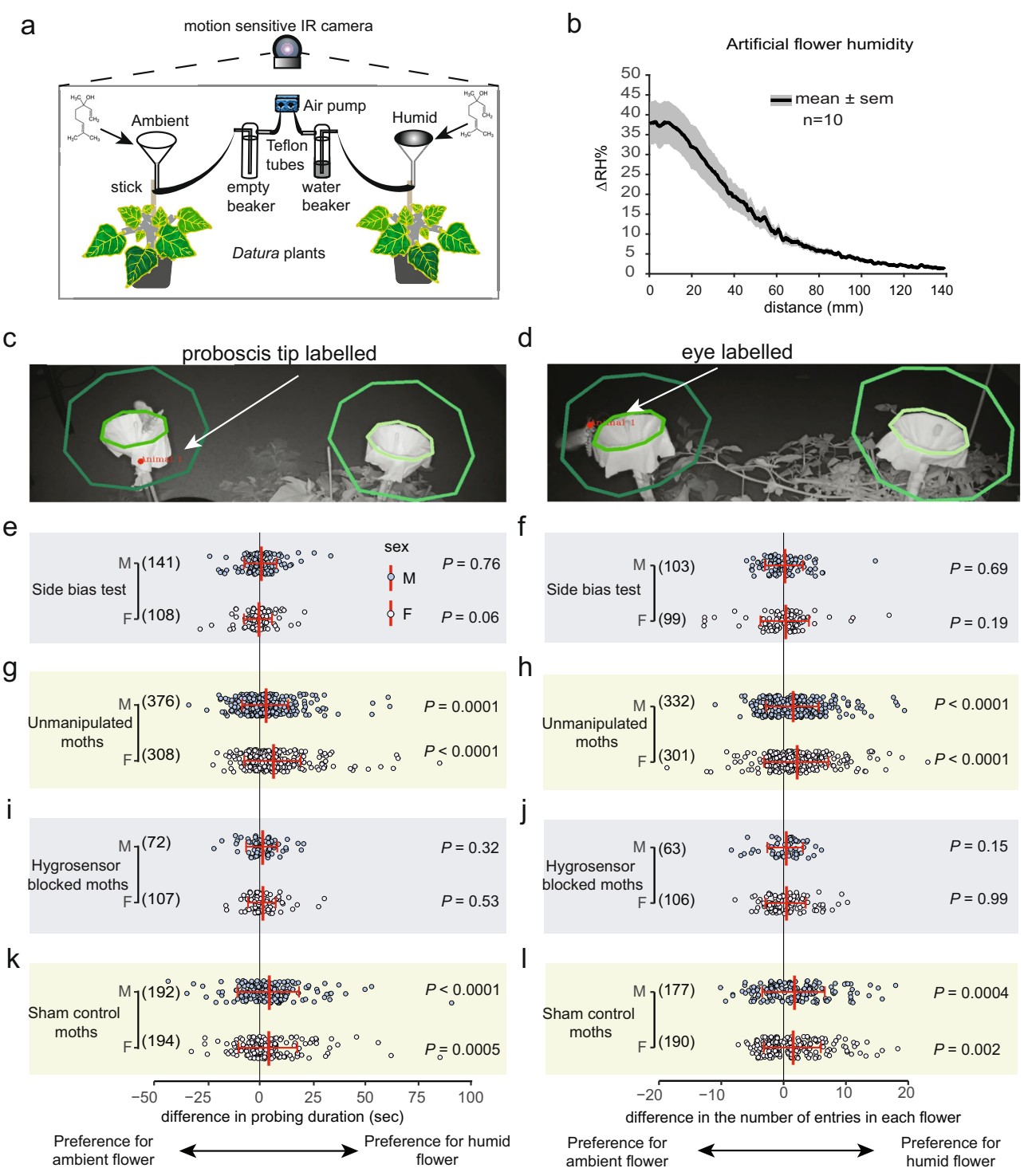

not completely override their innate preference for the humid flower by visiting the rewarding ambient flower more frequently. This intriguing result is analogous to the responses of a related hawkmoth, *Macroglossum stellatarum*, that shows a strong innate preference for blue flowers, which can be overridden, but only partially, by training moths to visit the less preferred yellow flowers[53].

To evaluate the fitness benefit for moths, we measured their flower handling time until proboscis contact with nectar was established in each treatment (see Methods). Accordingly, the increased probing and hovering times translate into more energy (Joules) expended by moths before nectar discovery (Fig. 5j–l). The time to

reach the nectary was shorter when the humid flower was paired with a reward than when only the ambient flower was paired with a reward (Kruskal–Wallis one-way ANOVA: $F = 6.97$, Df=2, $P = 0.0045$; Fig. 5j). Calculations show that moths expend significantly more energy to find nectar in the negative association treatment (Fig. 5j, k), in which the ambient flower is rewarding, but the humid flower is empty, suggesting that it is costly for moths to behave against innate preference. Since a fixed amount of nectar reward (200 μl) was offered, the total energy gained by moths from extracting the nectar from one flower equals 741 J. Figure 5k is identical to panel 5j because we input values from (j) in the formula and present it in a different currency. Figure 5l is the

**Fig. 4 | Moths are innately attracted to humid flowers in a binary-choice behavior assay. a** Setup for the two-choice behavior assay using artificial flowers mounted on two non-flowering *Datura* plants. Chemical structures indicate the addition of scent (bergamot oil) to both artificial flowers (funnels) at the start of the experiment. Air was pushed through the base of the funnels using an air pump (blue). For humid flowers, the air was pushed via Teflon tubes into a water beaker to generate saturated air, whereas, for ambient flowers, the air was pushed through an empty beaker. Overnight moth visits were video recorded using a motion-sensing IR camera (see Methods). **b** Measurement of the vertical gradient of floral humidity of the artificial humid flower used in the behavioral experiment. Data were shown for *n* = 10 transects as mean (solid line) ± SEM (gray shading). **c, d** Representative images from the videos stored by the motion-sensing camera show moths interacting with flowers. Circles around the flower show the region of interest we drew for position analysis of the moth proboscis (**c**) and head (**d**) during probing or entering flower headspace. **e–l** Behavioral responses of male (slate blue) and female (pale purple) naïve moths for the indicated treatments towards reward-less ambient and humid flowers except for the side-bias test where both flowers presented were ambient humidity. Dots show differences in the duration of proboscis contact and the number of entries in each flower. Red line plots show the mean ± SD. Numbers in parentheses indicate the number of videos in which the labeled body part appeared in the region of interest (**c, d**) for further analysis. The number of trials for each treatment was as follows: (**e, f**) Side-bias test *n* = 12 nights for both sexes; **g, h** Unmanipulated moths: *n* = 11 nights for both sexes; **i, j** Hygrosensor-blocked moths: *n* = 10 nights for both sexes; **k, l** Sham control moths: *n* = 12 nights for males, *n* = 13 nights for females. For each night of the experiment, 2–4 naïve moths were released in the behavior room. A two-tailed one-sample Wilcoxon signed-rank test against zero was performed on the data. Source data are provided as a Source Data file.

inverse of Fig. 5k because the net energy gain is the inverse of net energy expenditure; see Methods for calculations). Therefore, the net energy gain was equally high in the no-association and the positive association treatments, but significantly lower for the negative association treatment (Tukey's HSD: *P* = 0.0072; Fig. 5l). In summary, floral humidity facilitates nectar discovery when it reliably informs nectar presence, ultimately resulting in higher fitness benefits to moths via shorter handling time.

Taken together, the enhanced cost of the overstated floral humidity of *Datura* (Fig. 1), which exceeds that of a nectar evaporation cue by a power of ten (Supplementary Fig. 1c), combined with high signal efficacy (Fig. 1c, f), sensitive physiological responses to (Fig. 2) and persistent behavioral preference for floral humidity (Fig. 4), along with inferred fitness benefits of floral humidity to plants (senders) and foraging hawkmoths (receivers; Fig. 5j, k), strongly suggests a role of floral humidity as a signal, not a cue, in this nocturnal plant–pollinator interaction.

## Discussion

Flowers are often described as billboards or signposts, with reference to the information that they communicate to pollinators at different spatial scales. In this "advertising" context, conspicuous floral displays of scent and color can orient foraging pollinators from a distance but do not necessarily inform them about the presence of floral rewards once they arrive at the flower. In a previous study using *Oenothera cespitosa* flowers, we demonstrated that floral humidity resulting from nectar evaporation can serve as a cue for nectar presence to foraging pollinators at the flower's threshold[29]. Numerous studies outline how cues can evolve into signals if they benefit both senders and receivers and incur signaling costs that are offset by fitness benefits[22,23]. Senders must produce signals that are efficacious under most conditions and can be processed reliably by receivers[19,33]. Our study shows that floral humidity can be physiologically decoupled from nectar presence in the trumpet-shaped flowers of *Datura wrightii* and is present at a ten-fold greater intensity than that of a nectar evaporation cue (Supplementary Fig. 1c). Thus, floral humidity in *Datura* is not limited to nectar as a source, and selection can act on humidity independently to amplify its signaling function. Our results not only address several proximate questions regarding the efficacy and perception of floral humidity as a trait in pollination but also provide inferences for ultimate questions such as the functional role of floral humidity. Below, we discuss the proximate and ultimate evidence supporting the interpretation that floral humidity, like scent and color, can function as a signal in the *Datura-Manduca* pollination system, but is most spatially relevant at the threshold of the flower.

Regarding proximate mechanisms of floral humidity, little is known about its efficacy: the ability to attract pollinators in the face of background humidity, wind, and temporal dynamics[33,54]. The humidity gradients (ΔRH) of *Datura* flowers greatly exceed background noise (plant or leaf headspace) or the calibration errors of most measuring probes (±2% RH)[55]. The finding that humidity is replenished rapidly after a pollinator's visit (within 30 s, Fig. 1f), ensuring its presence for subsequent visitors, suggests that it is a persistent physiological signature, much like scent, rather than an ephemeral evaporation cue in this system[56]. Unlike floral scent, humidity is likely a local stimulus for pollinators because it decays rapidly from the boundary layers of the floral tube to an ambient RH that can vary enormously (Fig.1a; also see refs. 29,30). Such a local stimulus will be encountered by moths at a flower's threshold while hovering, rather than downwind of the flower, like scent[54]. The influx of air generated by the vortices shed by the hovering wings of hawkmoths may drive headspace humidity out of the flower and over the moth antennae, as has been demonstrated for olfactory stimuli[57,58]. Floral humidity is certainly experienced by moths upon entering flowers (Figs. 1f, 2f), and its presence prompted moths to visit artificial flowers irrespective of nectar presence (Figs. 4, 5). Conversely, a previous experiment revealed that *M. sexta* moths flying in a divided 2 m wind tunnel do not preferentially orient towards scented artificial flowers in a more humid airstream (vs. those in ambient humidity)[49]. The available evidence supports the conclusion that floral RH is a close-range attractant in the *Datura-Manduca* system.

We used a custom-built stimulus delivery setup (Fig. 2e) to reveal the mechanism by which a pollinator can perceive floral humidity. *Manduca sexta* possess at least two putative hygrosensing sensilla on each antennal segment: the coeloconic type B (peg-in-pit) and the styliform complex sensillum (large peg, Fig. 2a–c)[38,39]. Electrophysiological recordings showed antagonistic activity between the dry and moist sensing neurons (Fig. 2h), a stereotypical feature of hygro-thermo sensing sensilla[43–45,59,60]. Both moist and dry sensing neurons were more sensitive to the rate of change in RH than to instantaneous RH (Fig. 3i, k). Rates of change as low as −0.30% RH/sec and +0.23% RH/sec were sufficient to trigger spike frequency changes of 1 imp/sec in dry and moist sensing neurons, respectively, whereas, for instantaneous RH, the dry sensing neuron required −3.56% ΔRH and the moist sensing neuron required +1.24% ΔRH for a spike frequency change of 1 imp/sec in *Manduca* moths. These findings reveal that pollinator hygrosensory neurons respond to floral humidity in real-time as they enter the floral headspace. This range of sensitivity values is comparable to those reported for other pollinators like honeybees[61] and supports the observed behavioral preferences of the smaller hawkmoth *Hyles lineata*, which could differentiate artificial flowers presenting 4–8% ΔRH[29], and the evidence that bumblebees (*Bombus terristris*) can discriminate only 2% ΔRH[52]. In this context, the ΔRH of *Datura* flowers constitutes a super-normal sensory stimulus.

Regarding ultimate questions related to plant–pollinator communication, we now consider the potential signaling function(s) of floral humidity. In the field of animal communication, there is a widespread expectation that signal evolution involves costs to the sender, that exceed those (e.g., respiratory costs) associated with cues, especially for icon signals[22]. The mechanisms of floral humidity production

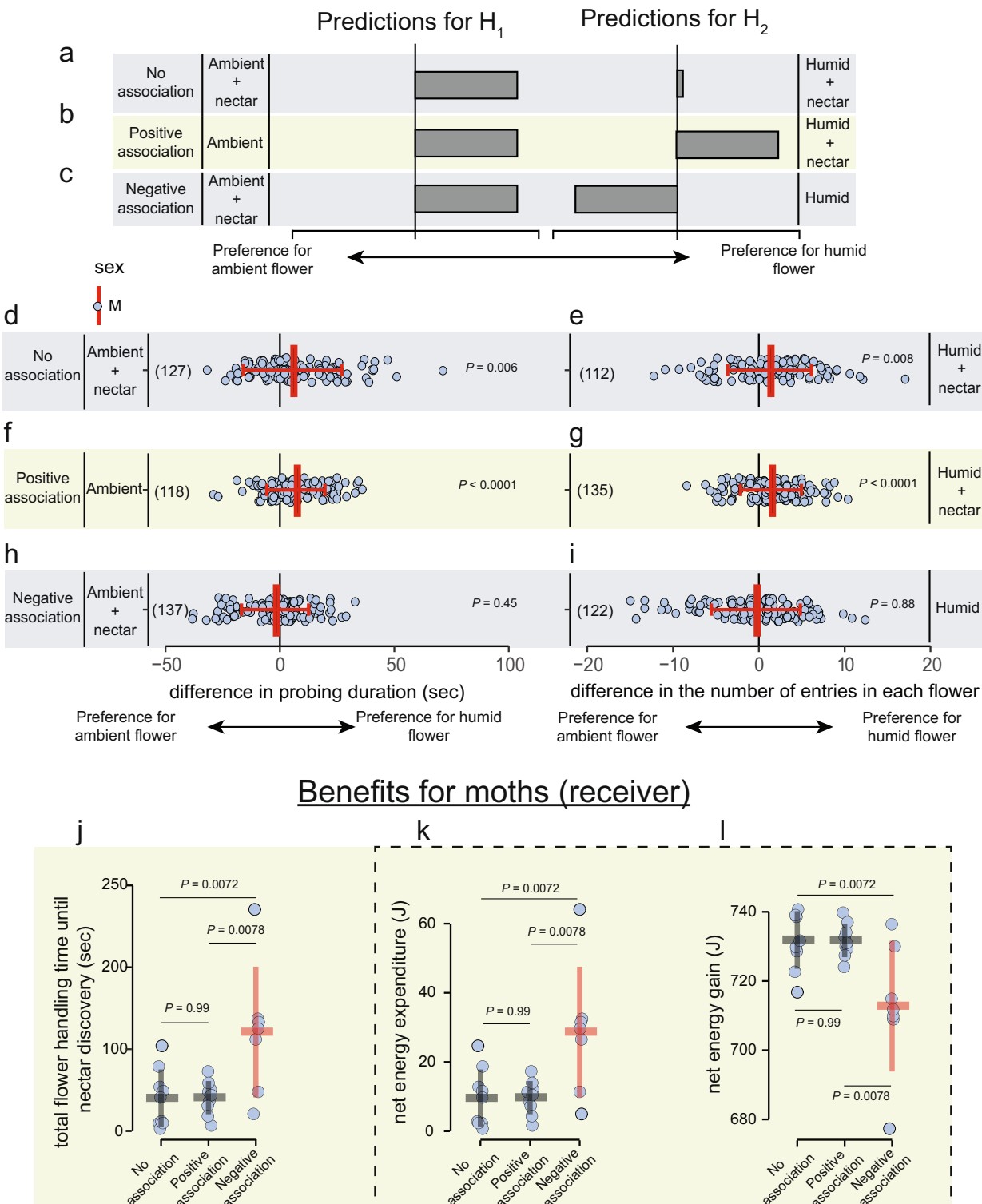

imply physiological costs with potential fitness consequences to the plant. For example, if floral humidity were an inadvertent nectar evaporation cue in *Datura*, then the intensity of ΔRH would be merely 3–4% (Supplementary Fig. 1c). The presence of stomates on the corolla suggests that *Datura* flowers likely incur costs to the plant, both in maintaining humidity gradients and floral turgidity through the night, in the xeric environments where they grow naturally[56,62]. In the case of

*Datura wrightii*, these costs may be especially high owing to the large surface area (164.49 ± 12.28 sq. cm, $n = 8$; Supplementary Table 8) and volume (83.95 ± 14.40 cu. cm, $n = 10$; Supplementary Table 8) of the flower. Measurements of fresh vs. dry flowers ($n = 6$) show that 87.40 ± 1.54% of the fresh floral mass is water, of which nectar mass accounts for only 2.18 ± 0.42% (Supplementary Table 9). Such an enormous water budget for flowers, along with nearly constant gas

**Fig. 5 | Moth responses to floral humidity decoupled from rewards reveal the signal-like role of floral humidity. a–c** Qualitative predictions outlined for hypotheses H$_1$ and H$_2$ for each treatment regarding moth preference for plants with humid and ambient flowers with the sugar rewards presented in either one or both flowers (see Table 1 for details). Panels show, **a** No-association treatment where both ambient and humid flowers provide nectar rewards, **b** Positive association treatment where the humid flower is rewarding, but the ambient flower is empty, and **c** Negative association treatment where the ambient flower is rewarding but the humid flower is empty. **d–i** Experimental results of the three treatments outlined above. Response variables are the same as in Fig. 4. Data show individual visits (light blue dots) with mean ± SD (red lines) for each indicated hypothesis. Note that only males were used in this experiment. Numbers in parentheses indicate the number of videos analyzed. The number of trials for each treatment was as follows—**d, e** No association: $n = 9$ nights; **f, g** Positive association: $n = 9$ nights; **h, i** Negative association: $n = 7$ nights. A two-tailed one-sample Wilcoxon signed-rank test against

zero was performed on the data and the $P$ values are mentioned next to each plot. **j–l** Analysis of moth interactions with flowers for each experimental treatment before establishing contact with the nectar reward. Panels show, **j** total flower handling time until nectar discovery, **k** Energy expenditure in Joules calculated from the hovering duration shown in (**j**), **l** Net energy gain calculated by subtracting the energy spent shown in (**k**) from the total energy obtained by consuming the offered sugar reward. The dashed box represents data derived from the handling time values shown in panel (**j**). Dot plots show values for individual nights and line plots show the mean ± SD for $n = 9$ each for the no association and the positive association treatment group, and $n = 7$ for the negative association treatment. Red line plot indicates a significantly different treatment group from the other two. $P$ values are shown on top for a two-tailed One-way ANOVA followed by Tukey HSD for comparisons between treatment groups. Source data are provided as a Source Data file.

**Table 1 | Experimental treatments decoupling floral humidity and nectar presence with alternative hypotheses and predicted effects on flower choice**

| Treatments | a. No-association | | b. Positive association | | c. Negative association | |
|---|---|---|---|---|---|---|
| Flowers with or without rewards | Ambient | Humid | Ambient | Humid | Ambient | Humid |
| | + | + | − | + | + | − |
| Information content of floral humidity | Humidity is uninformative for nectar presence | | Humidity is informative for a (+) association with nectar | | Humidity is informative for a (−) association with nectar | |
| H$_1$: Moths show a fixed preference for floral humidity; Predicted: | Prefers humid flower | | Prefers humid flower | | Prefers humid flower | |
| Observed: | Yes | | Yes | | No | |
| H2: Moth preference can be modified by experience; Predicted: | Loss of preference [see ref. 27] | | Reinforcement of preference | | Learns to prefer ambient [see ref. 104] | |
| Observed: | No | | Yes | | No | |

+ indicates the presence of sugar solution in the artificial flower, − indicates an empty flower.

exchange through corolla stomata, predicts that drought-stressed plants should produce fewer flowers. Indeed, *Datura wrightii* plants that are water-stressed produce shorter and fewer flowers that yield fewer viable seeds, at a direct cost to reproductive fitness[63,64]. These physiological costs, like the energetic demands or predation risks associated with vigorous courtship displays[65,66], imply that floral humidity must confer significant benefits in the currency of reproductive success. For *Datura*, those benefits accrue with repeated visitation by *Manduca* pollinators. Even in the absence of a nectar reward, moths showed perseverance by probing longer on the humid flower than on the ambient flower (Fig.4g). Moths also entered the humid flower more frequently while probing (Fig. 4h). Flower entry shows commitment to nectar-feeding, increasing the chances of pollen deposition and export, via dose-dependent pollen loading on moth body parts[50].

Plant–pollinator communication often is multimodal[67–69], including traits associated with primary metabolism. Apart from floral RH, moths also experience floral $CO_2$[25,27]. In our experiments, moth preference for humid flowers persisted throughout the night. This finding differs from moth preference for floral $CO_2$, which diminishes to chance level after the first choice, in the absence of differential reward[27]. The variable preference of moths towards floral primary metabolites is consistent with the temporal dynamics of $CO_2$ and humidity in *Datura* flowers. Floral $CO_2$ in *Datura* flowers dissipates to background levels within the first two hours after anthesis[25], whereas floral humidity remains consistently high throughout the evening, in parallel with floral scent and color (personal observation). The multimodal interplay between primary and secondary metabolites in shaping pollinator behavior is widespread[70] but is rarely addressed. Manipulative experimental studies of the cycad cones of *Macrozamia lucida* and their thrips pollinators[71] revealed that thrips are neutral to $CO_2$ but are repelled strongly by high temperature, humidity, and cone volatile production. Such studies reveal the temporal dynamics of

signals and cues in conjunction with pollinator behavior, advancing our knowledge of plant–pollinator communication beyond floral advertisement.

A longstanding question in the study of ecological communication is the degree to which signals should always benefit both sender and receiver, given prevalent conflicts of interest between them[23]. In plant–pollinator mutualisms, it is in the plant's interest to maximize pollination services for minimum cost, whereas pollinators aim to maximize fitness (energy gain) per flower visit[72]. The evidence presented here confirms that floral humidity is not an uninformative trait for pollinators, but also compels us to examine other possible functions and conditional outcomes. In our study, moths showed a strong sensory bias for humid flowers, whether sugar rewards were absent (Fig. 4) or present (Fig. 5d, e). Rewarding flowers could benefit by producing higher humidity to exploit pre-existing bias in moths, a phenomenon that is not uncommon in plants exploiting olfactory and visual biases in insects[73]. It is plausible that floral humidity may have non-pollinator functions such as preventing pollen desiccation[74], ensuring stigma receptivity[75], and promoting pollen tube growth[76]. We could imagine a scenario in which floral RH is an exploitable cue for nectar-robbing animals that puncture floral tubes, which consequently reduces plant reproductive success[77].

In *Datura wrightii*, floral humidity is physiologically decoupled from nectar, presenting potentially conflicting information to its hawkmoth pollinator *Manduca sexta*. To explore the potential for plant–pollinator conflict in the use of humidity gradients as a signal, we experimentally decoupled the presence of nectar with floral ΔRH in artificial flowers (Fig. 5). Thereby, we tested two alternative hypotheses concerning hawkmoth response, neither of which were completely supported by our experimental findings (Table 1). For H$_1$, in which moths show a fixed preference for humidity, we predicted that moths should simply prefer humid flowers irrespective of nectar reward, which would reflect a somewhat extreme (and potentially exploitable)

preference. Indeed, moths strongly preferred humid flowers whenever they were rewarding (Fig. 5d–g), or when neither humid nor ambient flowers were rewarding (Fig. 4). However, moths showed no preference in the "negative association" treatment, when the ambient flower was rewarding, but the humid flower was empty, presenting a conflict between the moth's innate bias and its expectation of a reward (Fig. 5c, h, i). For H$_2$, in which moth innate preference is modified by experience, we predicted that moths should lose preference for humid flowers in response to the uninformative "no association" treatment and should learn to prefer rewarding, ambient flowers in the "negative association" treatment (Table 1). Neither of these predictions were upheld (Fig. 5d–i), underscoring the need to test additional hypotheses addressing contextual or multimodal aspects not tested here.

These results suggest a potential avenue for plants to use floral RH as a deceptive signal to exploit pollinator preference. For example, rare (nectar-less) mutant *Datura* plants might benefit by cutting nectar costs while exploiting hawkmoth preference for humid flowers, if they remain equally attractive to moths (Fig. 5h, i). Although the humid flowers of *Datura* (and *Oenothera*) species generally offer copious nectar rewards to hawkmoth pollinators[78], there is always a potential for incomplete honesty where flowers could signal high humidity but present low volumes of nectar[79]. However, our findings predict that (food-deceptive) hawkmoth-pollinated plant species, which always lack nectar (e.g., *Plumeria rubra* and *Brassavola nodosa*), would benefit by adding humidity gradients to their visually conspicuous, fragrant flowers[80,81].

Returning to the criteria by which signals are defined[23], we have characterized the magnitude, spatial-temporal dynamics, efficacy, and physiological sources responsible for the production of a supernormal humidity gradient in the flowers of *Datura wrightii* as a sender, satisfying the first criterion. Conversely, we have shown that the receiver and primary pollinator, *Manduca sexta*, responds to changes in floral humidity through highly sensitive antennal sensilla and demonstrates an innate preference for humid flowers, satisfying the third criterion. Moths benefit by reducing their flower handling cost on humid flowers (Fig. 5j–l), whereas, from the plant perspective, multiple entries of moths into *Datura* flowers increase pollen loading[50], promote siring success[34], and cross-pollination[34,64], satisfying the fourth criterion. Indeed, reduced probing by *Manduca sexta* due to low nectar presence (and likely lower humidity) has been shown to lower seed set and impact plant fitness in related *Petunia* plants[82]. The functional role of floral humidity in this interaction fits best with that of a mutually beneficial signal, especially given that *Datura* nectar is refilled over time[25]. The only requirement not addressed in our study is the second criterion, which demands evidence that a signal "has evolved for the purpose of conveying information to the receiver"[23]. This is the most difficult criterion to satisfy without the benefit of trait-independent phylogenetic hypotheses for both sender and receiver lineages[19], onto which the kinds of mechanistic data outlined here (e.g., magnitude of floral humidity gradients, sensitivity of moth hygrosensory response) can be mapped for sister taxa of *Datura* plants and *Manduca* moths[83–85], a task that is beyond the scope of the present study. Such information would allow us to distinguish between alternative evolutionary pathways, such as the "signaler precursor route", in which signals originate from informative cues provided by the signaler and are honed by selection to convey information more effectively, vs. the "recipient precursor route", in which signalers evolve to match strong pre-existing receiver biases that arose under contexts independent of communication[22,23]. From the receiver end, we know that *Manduca sexta* moths live longer and imbibe higher sugar concentrations under humid conditions[86], prefer to fly towards more humid air in wind tunnel bioassays[49] and that a distantly related hawkmoth species (*Hyles lineata*) also shows an innate preference for modestly humid artificial flowers (ΔRH 5–6%) in the absence of nectar[29]. Thus, the innate preference of *M. sexta* moths for humid flowers connects to deeper physiological imperatives independent of flower foraging and may be plesiomorphic among hawkmoths.

Our findings suggest a broader set of roles for the significance of floral humidity to flower-visiting animals. One possibility is that floral humidity serves as an inviting stimulus prompting pollinators to enter a flower. This "microhabitat" hypothesis may also be relevant for the many smaller arthropods that utilize flowers for mating, breeding, and protection without necessarily benefiting the plant[87]. Another possibility is that floral humidity indicates either the presence or the location of nectar. A similar role has been implied for visual nectar guides in bumblebee-pollinated flowers[10]. Additionally, floral humidity may indicate larger nectar volumes and better nectar quality (sugars, amino acids, and micronutrients)[88] ensuring that, on average, pollinators benefit from attending to this signal. Maynard-Smith and Harper[22] suggest that signals whose information content is similar or related to the associated resource should be called "icons" whereas signals that are mechanistically removed from the resource (e.g., floral scent or color) would be regarded as "symbols". Hence, the enormous, sustained ΔRH in *Datura* flowers could be considered an icon of the plant's capacity to offer copious floral nectar in a severe desert environment. Our data most closely approximate this distinction because although ΔRH and nectar-standing crops are decoupled, RH may indicate the *Datura* plant's capacity to offer rich nectar rewards despite the high cost of water stress. Future studies should test whether variation in floral RH is, in fact, predictive of nectar-standing crops or secretion rates in the wild.

Finally, we suggest that floral humidity could have evolved as a communication channel between plants and pollinators in the earliest stages of plant–pollinator diffuse coevolution, in which brood sites or mating sites are proposed to have been more important than energetic rewards[89]. This notion that floral humidity might be an ancient communication channel in plant–pollinator interaction is in line with the evidence that Ionotropic receptor genes (IR) that mediate hygroreception[90–92] are conserved across arthropods and appear to have preceded the evolution of olfactory receptors[93]. Collectively, the efficacy of high floral RH in *Datura* as a trait, the salient behavioral and physiological responses that it evokes from moth pollinators, the non-trivial costs to the plant's water budget in a desert environment, and the fitness advantages accruing to both sender (*Datura*) and receiver (*Manduca*) reveal an unexpected signaling channel in plant–pollinator interactions.

## Methods

### Manduca sexta colony

We raised *Manduca sexta* from egg to adult in a laboratory walk-in growth chamber maintained at 24 °C and 50–60% RH with a 16:8 light: dark cycle. Caterpillars were fed a cornmeal-based artificial diet prepared in the lab. Late-stage caterpillars were transferred to individual cavities in wooden pupation blocks. After 7–10 days of pupation in the wooden blocks, pupae were transferred to the greenhouse to a moth breeding cage and left with a tomato plant for egg collection. Pupae used for the experiments were isolated from the lab breeding colony, separated by sex, and placed in 35 × 35 × 60 cm (BioQuip) cages until ready for experiments.

### *Datura wrightii* plants

*D. wrightii* seeds from Tucson, Arizona, USA, were requested from the seed bank at Radboud University, Nijmegen, Netherlands (Accession number: 944750169). Seeds were soaked in water for 24 h, followed by a rinse in 50/50 bleach water, and further soaked for 2 days in 0.1% Gibberellic acid. After soaking, seeds were nicked and placed on a wet filter paper in a petri dish until they germinated. Germinated seeds were sowed in 1-gallon plastic pots and placed in the greenhouse facility at Mudd Hall, Cornell University, under a 16:8 light: dark cycle. As necessary, plants were re-potted in a 3-gallon plastic pot, trimmed

as needed, and regularly watered with 21-5-20 fertilizer (nitrogen-phosphate-potash). Plants continued flowering throughout the year.

## Vertical and horizontal gradients of floral humidity

For floral humidity measurements, flowering *Datura* plants were brought to a laboratory room from the greenhouse after anthesis. Floral humidity transects were carried out during the first 2 h after anthesis at varying levels of background humidity throughout the year. The room temperature was 23 ± 2 °C and background RH varied from 12 to 68%. Typically, the background humidity was high in the summer months (40–60% RH) and low in the winter months (10–30% RH). In preparation for measuring vertical or horizontal humidity gradients, flowers were held straight using bamboo sticks and metal wires. The Omega, Inc. hygrosensor probe (model 314 A) was screw-fixed to a syringe pump (kd Scientific model 100) to move the sensor gradually but continuously in either vertical or horizontal transects (see ref. 29). The starting point for the vertical transects was the base of the flower tube, with the endpoint a few centimeters above the flower opening. At the start of the transect, the probe was lowered to the base of the corolla tube near the opening of the nectaries, ensuring that the probe head did not damage the anther filaments and the style. A vertical transect of 140 mm was carried out once for individual flowers with the probe moving at 0.21 mm/s. The slow speed ensured minimal mixing of air and allowed the hygrosensor to equilibrate. As with the vertical transects, horizontal transects were carried out 0.5 cm above the surface of the open flower across its diameter. The reference probe was placed 10–20 cm away from the flower at the same height as the flower opening. Output from the hygrosensors was compiled in the software provided by Omega. The humidity and temperature data were stored for every second of the transect, amounting to 650 points per transect. ΔRH was calculated by subtracting the ambient RH from floral RH, and the data were visualized in MATLAB 2019. Floral humidity data were collected as described here for all floral manipulations and artificial control flowers. We sampled floral headspace humidity in the tube and at the opening of *Datura* flowers growing in natural settings near Tucson, Pima Co, Arizona, USA. These settings included an experimental plot at Roger Road in urban Tucson, belonging to the Univ. of Arizona (32°16'41.3"N 110°56'18.5"W, 715 m), a piñon-juniper-boulder habitat at the upper elevational limits of the plant's distribution at Windy Point (32°22'07.0"N 110°43'00.8"W, 2013 m) in the Santa Catalina Mountains, and in natural grassland habitat in the Santa Rita Experimental Range (31°47'01.5"N 110°49'32.3"W, 1322 m). We measured the ambient humidity and temperature adjacent to the flower and noted the weather conditions at each location (Supplementary Table 3).

## Effect of breeze and nectar extraction on floral humidity

Conditions in nature are unequivocally more dynamic than in the laboratory. We expected the wind to reduce the boundary layers of the flower surface. To test that, we generated an artificial breeze of ~0.4 m/s over the flower using a clip-on fan (15 cm diameter) in a laboratory setting. A black air filter pad 50 × 100 × 0.5 cm was placed between the fan and the flower to reduce airspeed and create a laminar flow. The distance between the fan and the flower was roughly 15–20 cm. If the floral RH gradient is an outcome of passive nectar evaporation, we would expect nectar removal during hawkmoth visits to reduce floral RH[29]. We extracted floral nectar by using a 1 ml disposable syringe to pierce through the base of the nectar tube and remove nectar from the five individual nectaries of each *Datura* flower before initiating the transect. The experiments were carried out in a specific order. First, vertical transects were taken from control (unmanipulated) flowers in still air. Subsequently, a gentle breeze was applied to the flowers, and another transect was taken for the "breeze" treatment. Next, the nectar was extracted from the flowers, and the fan was turned off to measure

the humidity of the nectar-extracted flowers in still air. Lastly, the fan was turned on and another transect was taken for the "breeze + nectar extracted" treatment.

## Floral nectary and stomate blockage

Another way to test the contribution of nectar diffusion to floral RH is to occlude the floral nectar tubes (see ref. 29). Accordingly, each of the five individual nectaries of the *Datura* flower was blocked with petroleum jelly applied locally using a narrow tube connected to a syringe filled with the jelly. An alternative model to produce floral RH gradients is active gas exchange through floral stomata (a physiological mechanism), rather than (or complementary to) diffusion from nectar (a physical mechanism). To block the stomates of *Datura* flowers, petroleum jelly was smeared on the inner (adaxial) surface of the corolla, as shown in Fig. 1d (also see ref. 31). These experiments were carried out in the following order. First, vertical gradients of humidity were measured for the control (unmanipulated) flower. Next, the nectar was blocked, and another transect was taken. Finally, the inner surface of the flower was coated with jelly and a vertical transect was measured. In a separate experiment, the floral humidity of the control flower was followed by the humidity of a flower coated with the jelly on the outer (abaxial) surface of the corolla as a sham control for the use of petroleum jelly and its potential interaction with water vapor and flower health. There was no indication of flower damage from using the jelly. This was confirmed by leaving jelly-coated flowers on the plant overnight and visually comparing them with unmanipulated flowers the following morning.

## Stomatal counts

To account for the large surface area of the *Datura* flowers, its corolla was divided into four zones from the base of the flower tube (location 1) to the flower limb (location 4; see Fig. 1e). The corolla surface was peeled off by hand at these four locations to expose the thin epidermis on the inner surface. Epidermal peels were stained with dilute safranin for 15 s, rinsed in water, and mounted on a slide with a coverslip to visualize them at 20x under a Nikon Eclipse 80i compound microscope. Digital photos were taken of the prepared slides for peels at each location (Supplementary Fig. 5). Subsequently, to note the scale of the image, a picture was taken of a reference slide with a 1 mm grid engraved. Stomata were counted manually within a 1mm² area drawn on the images using image J.

## Simultaneous measurements of floral humidity and moth interactions

Fully opened flowers were excised from the plant, immediately placed in a conical beaker filled with water, and placed in a nylon mesh insect cage (BioQuip, Inc.; 71 × 71 × 122 cm) within a laboratory room. Only male moths were used in this experiment and were isolated from the lab colony on the day of eclosion. Moths were trained to visit and handle *Datura* flowers at least one night before the experiment was conducted. The SHT31-D hygrosensor (Adafruit) was used for this experiment. The hygrosensor was connected to an Arduino Uno that was connected to a computer and operated through a custom-written Matlab code. The sensor was programmed to collect 10 data points per sec (upper limit) and was left running to collect data for 3 min while a moth was introduced to the insect cage. The background RH was noted at the start of every trial. Light intensity in the room was 0.1 lux, measured using a light meter (Reed LX-1102). Moth visits were filmed using a Canon DSLR camera (EOS Rebel T6i) at 60 frames/sec for the 3-min duration the sensor recorded floral humidity for each trial. The video data were aligned with the floral humidity data using a custom Matlab script[94] (see Supplementary Video 1).

To measure the floral humidity experienced by moths as they enter and exit flowers, the hygrosensor was inserted in the flowers for 5–6 s and removed for ~10 s to mimic the behavior of moths when

introduced to a cage with a single *Datura* flower, based on actual visits of moths in our experiment.

## Floral surface area and volume measurements

For surface area measurements, $n = 8$ flowers were cut open and flattened for a couple of hours using a wooden herbarium press. Pictures were taken of each flattened flower with a scale reference next to it. The surface area was measured (sepals excluded) using Image J software (Supplementary Table 8).

For volume measurements, a separate set of $n = 10$ flowers were held upright, and water was added to the floral tube until it overflowed. The amount of water each flower could hold within its fused corolla was measured using graduated cylinders (Supplementary Table 8).

## Flower water budget measurements

Flowers were excised from the plants in the morning before they senesced. Nectar was extracted by slitting the bottom of the nectar tube near the ovaries and was collected in 1.5 ml Eppendorf tubes. Fresh nectar and flower weights were recorded. Flowers were then oven dried for 2 days at 50 °C to record their dry weights (Supplementary Table 9).

## Two-choice behavior assays

Two to four pupae of both sexes were isolated from the lab colony and placed in separate nylon mesh insect cages $40 \times 40 \times 60$ cm (BioQuip, Inc.) in the greenhouse. We used 4–5-day old, starved moths for the experiment. Moths were released in the experimental lab room $3 \times 6$ m (width × length) at dusk. Two potted non-flowering *Datura* plants were placed in the center of the room with two white funnels (Büchner funnels, 9 cm diameter) covered with a white paper towel attached to the plants, to mimic *Datura* flowers. The spectral reflectance of the paper towel matches closely with the authentic *Datura* flowers (Supplementary Fig. 7a). A small night light was plugged into the wall opposite the plants, and window blinds were closed, yielding light intensity less than 0.01 lux. The artificial flowers were attached to a bamboo stick and inserted within the *Datura* pots ~50 cm above the ground and 50 cm apart from each other. Either humid or ambient air was supplied to the base of the artificial flowers through Teflon tubes connected to an air pump with two outlets (Topfin AIR 4000). The ambient flower received air passed through an empty beaker, whereas the humid flower received air pushed through a water-filled beaker resulting in a 0.3–0.4 m/s airflow at both flower openings. A 2 ml syringe plunger was inserted inside the tube of the artificial flower to mimic the grooves of the authentic *Datura* flowers. A cotton-tipped swab was glued to the center of the plunger to occupy the space taken up by the stamens and style in an authentic flower (Supplementary Fig. 7b). Each evening, 5 µl bergamot oil was added to the cotton swabs of both artificial flowers to provide a standardized, surrogate floral scent. A motion-sensitive IR video camera (Amcrest IP3M-941B) was placed to film overnight moth visits to both flowers. Videos were downloaded the following morning from the micro-SD card and were saved on a hard drive under appropriate treatment folders. The *Datura* plants were replaced and cycled through the ten plants that were available in the greenhouse. Care was taken that plants had no blooming flowers during the trials. The positions of the ambient and humid flowers were alternated every night of the experiment.

For the rewarded assays, we used only flower-naïve male moths to exclude the oviposition context associated with female moths. The artificial flower was modified by attaching four pipette tips at the edges of the syringe plunger to create four nectary grooves in the artificial flower (Supplementary Fig. 7c). Individual pipette tips were filled with ~50 µl of 22% sucrose solution, amounting to 200 µl in each flower (upper limit of nectar offered by the *D. wrightii* flowers in greenhouse conditions), only once at the start of the experiment. Depending on the treatment, either one or both flowers were provided with a 22% sucrose reward. The following morning, nectary tubes were checked for consumption of sugar rewards, washed, and dried before using them for another trial.

## Behavior video tracking and analysis

We used an animal-pose tracking software SLEAP[94,95] to track pre-selected body parts on the moth to collect information on moth position and behavioral choice in the two-choice behavioral assays. We selected the eye, head, proboscis tip, thorax, wingtips, and abdominal tip of the moth's body for tracking-based analyses on videos obtained by the motion-sensing camera (Supplementary Fig. 8a). We labeled 1669 frames across 137 videos representing different behavioral trial sessions. A neural network was trained using SLEAP[95] v1.1.5 installed on a PC equipped with a Geforce RTX 2080 Ti graphics card. The network was trained for 83 epochs or until the network loss value plateaued. We qualitatively assessed the network prediction accuracy from tracked behavioral videos prior to applying the network to track all remaining behavioral videos. To derive metrics of the moth's body positions around the artificial flowers, we used SimBA[96] to analyze the positional output data from SLEAP for all tracked body points. Five regions of interest were drawn across each video to enclose the cup of each artificial flower, the space surrounding the flowers, and the entire frame (Supplementary Fig. 8b). For the probing duration, the proboscis tip labels were used, whereas, for the number of entries made into each region of interest, the eye was used for its proximity to the moth's antenna and thus the hygrosensing sensilla. The output files were subsequently analyzed in R (v.4.1.1). Videos with tracking anomalies (<5%) were hand-corrected and the data were entered manually.

For behavioral response analysis in the experiment with the sugar-rewarded flowers (Fig. 5), the videos were grouped into two categories: "before" and "after" nectar discovery. The handling time on either flower was scored manually by noting the time until the moths discovered the reward. Once moths contact the nectar with their proboscis after landing on the artificial flowers, they show a stereotypical behavior: they cease fluttering their wings and remain perched on the flowers with their proboscis extended in the floral tube for a prolonged duration to consume the reward. The cumulative handling time per night was the sum of all the probing duration of the moths on either flower until nectar was discovered. Moths did not find the nectar reward during 2 out of 9 nights in the "negative association" treatment, and during 1 out of 10 nights each for the "positive association" and the "no-association" treatment. For the behavior analysis of moths "after" nectar was discovered, we examined the same response variables: "probing duration" and the "number of entries" as for the empty flower assay (Fig. 4).

## Net energy calculations

To calculate total energy expenditure, we used the formula:

$$energy(J) = Power\left(W\,kg^{-1}\right) \times Time(sec) \qquad (1)$$

For power input values ($W\,kg^{-1}$), we referred to Casey (1976)[97], whose data show that a $1.2 \pm 0.08$ g moth (mean ± SEM) requires $0.237 \pm 0.01$ W g$^{-1}$ power to hover. For time values, we plugged in the total flower handling time (sec) for moths until proboscis contact with nectar was established.

To calculate net energy gain for moths, we subtracted the energy expended (J) in handling flowers from the energy gained (J) from consuming 200 µl of the *Datura* nectar mimic 22% sugar solution (recipe: 6.76 mg glucose, 5.54 mg fructose, 32 mg sucrose)[98] with a total sugar content of 44.3 mg per 200 µl solution. Therefore, the energy gain would be $0.0443\,g \times 4\,kcal = 0.1772$ kcal/flower (1 g sugar = 4 kcal), which converts to 741.4 J, because 1 kcal = 4184 J.

## Occlusion of the hygrosensing sensillum and sham control

Three day old moths were cold-anesthetized in a −20 °C freezer for 10 min. Once anesthetized, moths were viewed under a dissection microscope ventral side up and dorsal side placed over a cold metal block. A UV light-activated glue (Riverruns) was used to occlude the hygrosensors on the moth antennae (Supplementary Fig. 6b, d). The glue bottle opening was attached with a 20–200 µl pipette tip for localized application of the glue. The glue was applied along the leading edge of the entire antenna, completely coating the styliform sensilla. The glue was hardened under a handheld UV flashlight for 1–2 min. For sham control, only 5–10 segments at the base of each antenna were coated with the glue (beyond the scape-pedicel joint), leaving the rest intact. After the occlusion, moths were returned to the greenhouse for 1–2 days to recover from the handling stress, before using them for the behavior experiment. The morning after the trial, moths were inspected under the microscope to evaluate the coverage of the glue on the antennae. In some cases, moths were able to remove the glue, presumably while cleaning their antennae. Trials performed with such moths were excluded from further analysis.

## Humidity stimulus delivery setup

We used two air pumps (Uniclife UL25 Air Pump) to continuously push air at a flow rate of 2 L/min. The air was bubbled through an air stone immersed in water at room temperature. The resulting air saturated with water vapor served as input to two dewpoint generators (DG-4 DewPoint Generator, Sable Systems International). The dewpoint generators were set to operate in relative humidity controller mode, so that they outputted air at a fixed relative humidity of RH1 = 11% and RH2 = 90% corresponding to the temperature of the area adjacent to the moth antennae, $T_{amb}$, which was measured using a thermistor probe connected to the dewpoint generator. The airstream outlet of the dewpoint generator was fitted with a needle valve (Stainless steel High flow metering valve, Swagelok Inc.) with its head attached to a stepper motor (28BYJ-48 ULN2003 5 V Stepper Motor) controlled using Arduino Uno (Arduino Inc.) and a stepper motor driver board (5 V Stepper Motor ULN2003 Driver Board). We used a program written in Arduino IDE (open source) for regulating the airflow rate by controlling the valve opening position, motor speed, and motor acceleration. The valves were regulated such that the air outlets from the two dewpoint generators were antiphase as shown (see Fig. 2e). The air streams mixed and passed through a T-junction connector and delivered locally with an airspeed of 0.5–1.3 m/s at the recording site on the moth antenna. We placed temperature and humidity sensors (AdaFruit SHT31-D) near the antennae (within less than ~2 cm) to simultaneously measure the temperature and humidity of the delivered airstream. This apparatus allowed us to control the rate of the sinusoidal humidity stimulus as well as offer stationery or step-like stimuli. We partially programmed the humidity stimulus through MATLAB and viewed it as a real-time MATLAB figure simultaneously with the instantaneous electrophysiology output[94].

## Single sensillum recordings and spike sorting

We immobilized 2–3-day old moths in a 15 ml falcon tube. The falcon tube base was cut off just enough for the moth's head and antennae to protrude. The head of the moth was prevented from moving by fixing it to the tube base with a collar of dental wax. The moth's proboscis was extended and fixed to the tube with more dental wax to prevent proboscis movement from interfering with the electrodes. Moths were placed ventral side up under the microscope on a 10 × 10 × 2 cm (l × w × h) plexiglass block. The antenna was adhered to the plexiglass with a hand putty (Blu Tack). The styliform sensillum was viewed under a microscope (WILD M3C, Heerbrugg, Switzerland) at a 40x zoom objective attached with a 1x magnifying lens and 20x eyepiece. For single sensillum electrophysiology, a sharp 2.5 cm wax-coated

tungsten microelectrode (MicroProbes) with 2.5 MΩ resistance and a similar reference electrode was attached to a headstage (A-M systems, model 1800) fixed on a micromanipulator (Narishige). All electrical components of the electrophysiology rig were grounded to a wire in the room away from the rig. The wires attached to the hygrosensor placed next to the moth antenna were wrapped in aluminum foil to reduce electrical noise. The recording electrode was connected to a two-channel high impedance amplifier (A-M systems model 1800), the signal was bandpass filtered for 300 to 1000 Hz, a notch filter was turned on and the signal was transferred to a data acquisition device (National Instruments, Inc., model USB-6211). The data acquisition device was connected to a computer and the signal was visualized using the open-source software Spike Hound v1.2[99]. The sampling rate was set at 20,000 Hz and the individual recording sessions of 3–6 min each, were saved on the computer until further analysis. Selected raw electrophysiology traces were spike-sorted using an open-source Waveclus 3.0 toolbox[100]. The stimulus and the spikes were aligned, analyzed, and visualized in MATLAB (R2019a) using a custom script[94].

The micromanipulator was advanced to insert the recording electrode at the base of the styliform sensillum. The reference electrode, attached to an electrode holder and controlled by another micromanipulator, was inserted a few segments toward the proximal end of the same antenna as that of the recording electrode. Electrophysiology was performed on moths of both sexes, and a new moth was used for every new recording event. *M. sexta* antennae consist of one styliform sensillum on each segment of the antenna; thus, during one recording event, we attempted multiple sensilla on 5–10 segments of the middle portion of the moth antenna. The styliform sensillum is a complex of three to five individual sensing organs (papillae) located at the tip of the peg[40]. Therefore, a high density of cell bodies is present beneath the sensillum (see Fig. 2d). Many recording events pick up more than one unit of one cell type. A moist neuron was identified if it responded with increasing firing frequency when the stimulus humidity increased, whereas a dry sensing neuron was identified if it showed increased impulse frequency when the stimulus humidity decreased[45]. Therefore, the moist and dry sensing neurons are antagonistic to each other in their responses to changes in humidity. These two neurons are associated with a third neuron, the cold cell[101], that responds with high impulse frequency when the air temperature decreases but ceases firing when the air temperature increases. We did not analyze the responses of the cold sensing neuron in this study because the floral temperature was not different from the ambient. For every new recording event, the amplitudes of the dry and moist sensing neurons vary depending on where the tip of the electrode is in relation to the cell bodies of the neurons. However, the ratio of amplitude stayed constant throughout the length of the recording. We were able to record anywhere from a few minutes to a couple of hours from the cells within a sensillum. Males and females showed identical responses.

## Scanning electron microscope images

SEM images were taken of air-dried antennal samples of both sexes under a Zeiss Gemini 500 electron microscope. Samples were sputter-coated with gold for 30 s and imaged at EHT between 0.3 to 1 kV and WD between 2.4 to 7.7 mm.

## Statistics

All floral humidity curves were plotted using the shadedErrorBar[102] function in matlab. To evaluate statistical differences among floral humidity curves, we used R v.4.1.1 "nlme" package to fit nonlinear models to the data[103]. We started with a simple nonlinear mixed effect model with no effect of the different treatments and no random effect of the individual flowers on the model parameters. However, adding the effect of the different treatments in the fixed effects and the random effect of individual flowers significantly improved the model and lowered the AIC

value. Our final fitted nonlinear mixed effect model is as follows:

$$\Delta RH_i = y0_i e^{-\alpha_i \text{distance}} \tag{2}$$

The best-fitted model suggests that ΔRH (%RH above ambient) varies by treatment $i$ and decays exponentially from the initial value $y0$ for that treatment, to the final value at a decay rate of $\alpha$ by distance. The model allows for separate intercepts and decay rate for each treatment $i$ and includes random effects of individual flower transects on the intercept $y0$ and the decay rate $\alpha$. Using package "emmeans" we calculated the estimated marginal means and 95% confidence intervals for $y0$ and $\alpha$ for each treatment. We performed pairwise $t$-tests with post hoc Tukey adjustments to the $p$ values in comparing the $y0$ and $\alpha$ values between multiple treatments.

For the stomatal counts, we performed a Kruskal–Wallis test across the four locations at which we counted stomatal density. For all the behavior data, we performed either one-sample t-tests or Wilcoxon tests, depending on the distribution of the data (normal vs. not normal), with the null hypothesis being that the differences in probing duration and the number of entries between humid and ambient flowers are not different from zero. In other words, the null prediction is that moths cannot distinguish between humid and ambient flowers and visit both flowers equally. For comparisons between the flower handling time and energetics (Fig. 5j–l), we used a one-way ANOVA followed by a Tukey HSD post hoc test for comparisons between treatments.

We used MATLAB R2019b to generate the 3D scatterplots of the dry and moist neuron impulse frequency (y-axis) plotted against instantaneous RH (x-axis), and rate of change of RH (z-axis) (Fig. 3). The MATLAB curve fitting app, cftool was used to fit the three-dimensional polynomial linear regressions to the data of the form:

$$F = a + b\Delta RH/\Delta T + cRH \tag{3}$$

where $F$ is the impulse frequency of the dry or the moist neuron, $a$ is the height of the regression plane, $b$ is the slope for the rate of change in RH, and $c$ is the slope for instantaneous RH.

### Reporting summary

Further information on research design is available in the Nature Portfolio Reporting Summary linked to this article.

## Data availability

All data generated in this study are provided in the Supplementary information and as a Source Data file Source data are provided with this paper.

## Code availability

Custom codes used for data analysis are available on GitHub and linked to Zenodo: https://doi.org/10.5281/zenodo.7320037 (ref. 94).

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

## Acknowledgements

We are grateful to Jack Bradbury and Sandy Vehrencamp for discussions on signals and cues, which significantly improved the narrative of the manuscript. We thank John Putnam for greenhouse care, Deidra Jacobsen for help germinating *Datura* seeds, and Cole Ortiz for help with stomatal counts. We are grateful to Ron Hoy and Gil Menda for providing the electrophysiology equipment and Bruce Johnson for teaching electrophysiology to A.D. We thank Goggy Davidowitz and Judith Bronstein for hosting A.D. and R.A.R. in Tucson and for field assistance in measuring floral humidity. We are thankful to Gordon Smith and Shayla Salzman for their insightful comments on earlier versions of the manuscript. We are grateful to Joe Fetcho for access to the cryotome. Lynn Marie Johnson from the Cornell Statistical consulting unit provided help on the nonlinear models of floral humidity curves. This work made use of the Cornell Center for Materials Research Shared Facilities, which are supported through the NSF MRSEC program (DMR-1719875). Parts of this research were funded through Cornell Sigma Xi, Cornell CALS alumni grant, Cornell Neurobiology and Behavior dept. grant awarded to A.D. The neurophysiological experiments were supported through Cornell Neurotech Mong fellowship awarded to A.D. and P.J. in the labs of R.A.R. and A.D.S. A.D. is thankful for the F. Arthur and Jean Fenton Potter fellowship.

## Author contributions

A.D. and R.A.R. conceived the study, A.D. performed all experiments and analyzed data with help from P.J., C.C.V., and W.K. P.J. and A.D.S. designed the stimulus delivery setup. A.D. and R.A.R. wrote the paper; all authors edited the manuscript.

## Competing interests

The authors declare no competing interests.
