## [Peer Review File · Nature Communications]

A signal-like role for floral humidity in a nocturnal pollination systemReviewers' Comments:

Reviewer #1:

Remarks to the Author:

The submitted ms by Dahake and colleagues concerns the role of floral humidity in the *Datura* – hawkmoth pollination system. The authors show that *Datura* flowers release a quite considerable relative humidity gradient, which is not simply a result of nectar evaporation, but a consequence of gas exchange through specific pores in the stomata. Next, the authors examined how hawkmoth pollinators detect humidity cues. They find a subset of specialized on the antennae, seemingly housing the “canonical” wet, dry, and cold neurons that are typically found in insect hygroscensilla across insects. Electrophysiological recordings from these neurons also confirmed a hygrosensory function, and . Then the authors conducted a number of carefully controlled and thoughtful experiments aimed at deciphering the function of the emitted humidity, and found that the moths are innately attracted to the increased humidity, preferring humid, but nectarless flowers to nectar rewarding, but non-humid flowers even after repeat visits. The authors hence argue that floral humidity is an attractive signal in itself, and not just a cue used by the insects to locate nectar.

All in all, there is much to like here. The topic is an interesting, but poorly understood one; from the perspective of plants and pollination as well from the insect side. Humidity as signal and as cue for insects is likely important All experiments are carried out with great care and much thought. The text, perhaps a bit on the long side in places, is crafted with equal care to detail, as are the figures. I am hard pressed to find any flaws, or the need for additional experiments that would substantially improve the manuscript. That said, showing that humid flowers indeed export more pollen would have been nice. It is a bit odd that this desert dwelling plant seems to invest considerable resources (water) in generating this signal, which clearly only works at a very short distance. Is it possible that the humidity is a reward in itself? The risk of dehydration is an issue for most insects, and presumably also for *Manduca*. As such, the humidity would be an honest signal. As for humidity being an ancient signal. The IRs are indeed of old age, but whether the hygrosensory IRs are as well is uncertain. Moreover, it remains to be shown what the precise function in hygrosensation these IRs have. They are necessary for proper hygrosensation, but are likely not the actual receptors.

Very minor comment:

Line 242, wrong figure reference...

Reviewer #2:

Remarks to the Author:

My curiosity was piqued by the abstract of this manuscript when I read that it was about relative humidity in the floral headspace being perceived by hawkmoths and furthermore, that hawkmoths preferred more humid headspaces. Until then, I had not known that humidity could be used as such by pollinators. I was therefore surprised upon reading the manuscript that this had been documented before and so my main question was to find out what was novel about this paper. Upon reading further, I found that the main angle was in finding out whether floral humidity is a floral cue or a floral signal. The differences between these (according to this manuscript) is that floral cues usually benefit just one party and typically are not costly to produce, whereas floral signals usually benefit both parties and typically incur some cost to the producer. For example, floral pigments may be regarded as a signal because they increase plant fitness and also improve pollinator foraging efficiency. They are also costly to produce. The authors suggest that recent literature views floral humidity as a cue, although I can see that there are potential arguments for or against whether it benefits pollinators. While the authors argue that floral humidity is more likely a signal than a cue, I am not sure that they really demonstrate this adequately

At this stage, I would like to say that the authors did perform some truly beautiful and elegant experiments. They were wonderfully innovative and used a host of interesting, disparate, multi-disciplinary techniques to test their hypotheses. If ever there was a case to publish a paper just for the artfulness of the experiments – this would be it.

However, weighing up against the beauty of the experiments is the importance and novelty of the question and also whether the question was properly tested. Unfortunately, I could not get excited about this aspect of the manuscript: I found the distinction between signals and cues somewhat semantic and niche, but also very poorly developed. As a pollination biologist, I have never really seen the need to distinguish between them and the authors do not provide great reason why we should. I initially thought that this kind of information was lacking because the authors had tried to fit too much information into a small amount of space, but in fact, the manuscript is quite long. There are 5 multi-panel figures (one of them has 13 panels and sub-panels, another has 12). I think there is a general need for the authors to think more carefully about what they want the main angle of the paper is and how best to show/test it, as many of the things done in this paper (although cool) did not link up easily with the question at hand.

I was a little frustrated that as I came to the end of the introduction, there was not any sign of an hypothesis. I later found hypotheses embedded in various tables, figure legends and results. I find that having some broad, clear hypotheses in the introduction is important because it prepares the reader for the results, making them easier to interpret. What I found was a manuscript that was not easy to follow. I think this was often because it was seldom clear how the research at hand related to the broad question. Just as an example: why were the authors looking for vertical humidity gradients? How does this relate to the main question (cue or signal)? I think it may be very helpful to structure the headings and subheadings as questions. However, it is also important for the reader to know why each question is important. For example, why is it important to know why wing fanning can influence floral humidity?

Hypothesis testing: The main angle of the paper was that the authors try to distinguish between whether relative humidity is a cue or a signal. However, I found that the hypotheses were not clear enough to do this and I also do not think that the tests were appropriate. The 2 alternatives (cue versus signal) were briefly defined as follows: Cues are when only 1 partner benefits and when the trait is not costly to produce, while signals are when both partners benefit and when the trait is costly to produce. In fact, when you break it down, there are also categories which potentially don't count as a cue or a signal (e.g. when there is a cost, but only 1 partner benefits), and it is possible that RH falls into one of those un-named categories (see table 1). While I have tried to visualize the different categories in Table 1, this visualization may be quite simplistic because there may be instances when one partner is actively harmed by a signal (but perhaps this counts as no-benefit). Be that as it may, this suggests that there are at least 3 components to distinguishing between these classifications: 1) showing whether partner 1 benefits, 2) showing whether partner 2 benefits, 3) showing whether there is a cost. I question whether the authors do a good job of testing any of these. I could be convinced that the authors test whether RH benefits flowers – here the authors show that humid flowers receive more probes and are visited for a longer duration than less humid flowers. However, whether this has a positive effect on any real measures of fitness (e.g. seed production or pollen transfer) is an open question. In the wild, it is also questionable whether a moth will pass up the opportunity to visit a less humid flower once it gets close enough to assess humidity. The next part of the question is admittedly much more difficult to test: is there a benefit to the moth? Here, I cannot see any clear answer, and I do not think that the authors formally test this in any way. It is possible that using RH, moths are able to navigate and manipulate the flowers more easily, hence increasing their foraging efficiency and providing a benefit to the moths. Alternatively, the plants may be exploiting a sensory bias of the moths and tricking them into visiting flowers with little nectar, potentially reducing moth foraging efficiency. This seems like a real possibility since nectar and humidity do not seem to be coupled and moths appear to prefer humid flowers over less humid flowers, even if the less humid flowers have more nectar. The third part of the question (is humidity production costly?) is not a simple question either and I would argue that everything on a plant or produced by a plant is costly as it takes ATP to produce. From this point of view, I do not think that the definitions of signals versus cues have been particularly well thought-out. Perhaps a better question is whether the trait evolved in response to the other partner for communication purposes or whether the trait evolved for a different function (other than communication) and is simply being used by the other partner. My guess is that RH in floral tubes may be important because it reduces the evaporation of nectar or prevents flowers from wilting. I doubt that it evolved for pollinator communication. But I may be wrong – the authors have not tested this. But these questions expose the fact that the main angle of this manuscript (distinguishing between signals and cues has not been properly thought through). This is exemplified by the statement in L94: “The question of whether floral humidity may also function as a signal (like floral color and scent) hinges upon whether RH enhances floral fitness.” But this completely ignores the other 2 components of distinguishing signals from cues (see table 1).

Table 1. The components of trying to classify floral traits as signals or cues

Cost	Partner 1	Partner 2	Classification
Yes	Benefit	Benefit	Signal
No	No-benefit	Benefit	Cue
No	Benefit	Benefit	?
Yes	No-Benefit	Benefit	?

Lastly, contributing to the difficulty in reading this paper, I often found the writing in the manuscript quite awkward. Below, I highlight some examples of this from the introduction:

L42: “The recent expansion of studies in plant-pollinator interactions reflects increased recognition of their importance to ecosystem services, agricultural food security and biological diversity...” I found that this opening paragraph did not provide a strong hook to catch the reader. This paragraph did not make me want to read further and engage with the manuscript, it just felt as though the authors were trying to squash their interesting research into some applied box. It stands isolated and apart from the rest of the manuscript because it is.

L43: “Examining plant-pollinator interactions through a neuroethological lens presents unique opportunities to investigate the sender-receiver elements of ecological communication.” The authors need to be aware that their use of language is not easily accessible to readers. This sentence, which is full of jargon is a case in point.

L47: “Plant-pollinator interactions are maintained through floral signals and the innate responses they evoke in the target pollinator.” Is this not over-stressing the role of innate responses? Surely learned responses can also be very powerful. – think mimicry.

L57 Shouldn't this be “relative humidity (RH)”

L80 In a lab essay

L81 non-rewarding rather than unrewarded

L82 “pollinators ...can associate both dry and humid flowers with nectar presence under varied levels of background humidity, thus highlighting their flexible valence to floral humidity.” Again, this language is not easily accessible

L87 “Harrap and Rands³⁵ followed up their garden survey with manipulative experiments on two of their most humid flowers: *Calystegia sylvatica* and *Escholtzia californica*, yielding two important insights. First, funnel shaped morning glory flowers produced the highest Δ RH values (~3.7%) indicating that floral shapes that enclose the headspace can retain higher humidity levels³¹.” The authors are maybe not being critical enough of the work they cite: by comparing the RA of 2 flowers, one can't say definitively that it is floral shape which is influencing RA. It could be any other unmeasured trait.

L94: “The question of whether floral humidity may also function as a signal (like floral color and scent) hinges upon whether RH enhances floral fitness” Here I suddenly feel a little unsure about the distinctions between signals (both benefit, costly) and cues (one benefits, not costly). Surely most floral traits that are attractive to pollinators should benefit both the plant and the pollinator? In this case CO₂ should be regarded as a signal because it benefits both. As should humidity because others have demonstrated that it is attractive to pollinators. You may point out that there is a

difference in costliness. So what if it benefits both but is not costly? Or if it benefits one but is costly. For example floral scents or colors in mimetic flowers are costly and only benefit the plant. I do not feel that the differences between cues and signals in plants is properly covered or that it is important. Or that this manuscript actually helps us to distinguish whether humidity is a signal or a cue (mostly because RH does not fit properly into either of the two definitions given).

L102: why would evaporation make RA an ephemeral cue? Nectar is often constantly replenished. I am also unsure that nectar removal would immediately make floral tubes suddenly not-humid. I expect that humidity would remain for a long time after nectar removal (even if it was the nectar that produced the humidity in the first place).

L107: "Furthermore, their attraction towards humid flowers cannot be abolished by experimentally rewarding only the ambient flowers, indicating a receiver-bias towards floral RH." While I understand what the authors are trying to say in this sentence, I find the wording very awkward.

Response to reviewer #1

The submitted ms by Dahake and colleagues concerns the role of floral humidity in the *Datura* – hawkmoth pollination system. The authors show that *Datura* flowers release a quite considerable relative humidity gradient, which is not simply a result of nectar evaporation, but a consequence of gas exchange through specific pores in the stomata. Next, the authors examined how hawkmoth pollinators detect humidity cues. They find a subset of specialized neurons on the antennae, seemingly housing the “canonical” wet, dry, and cold neurons that are typically found in insect hygrosensilla across insects. Electrophysiological recordings from these neurons also confirmed a hygrosensory function, and .

There is some text missing here in our version of the reviews, but at this point the reviewer is summarizing our findings, rather than making a query...

Then the authors conducted a number of carefully controlled and thoughtful experiments aimed at deciphering the function of the emitted humidity and found that the moths are innately attracted to the increased humidity, preferring humid but nectarless flowers to nectar rewarding, but non-humid flowers even after repeat visits. The authors hence argue that floral humidity is an attractive signal in itself, and not just a cue used by the insects to locate nectar.

Yes, at least in the *Datura* model system studied!

All in all, there is much to like here. The topic is an interesting, but poorly understood one; from the perspective of plants and pollination as well from the insect side. Humidity as signal and as cue for insects is likely important. All experiments are carried out with great care and much thought.

We greatly appreciate the reviewer’s comments here.

The text, perhaps a bit on the long side in places, is crafted with equal care to detail, as are the figures. I am hard pressed to find any flaws, or the need for additional experiments that would substantially improve the manuscript. That said, showing that humid flowers indeed export more pollen would have been nice.

Many thanks – we have tried to edit the revised version to be more concise wherever possible, although we needed to add some text and new data to address other reviewer concerns. Regarding pollen export, we now draw a connection between our finding that moths spend more time entering and probing humid flowers and a parallel study performed in our own laboratory, which quantifies pollen removal per visit by the same *Manduca sexta* moths from the same *Datura wrightii* flowers used in our study. This connection is made explicit in the revised manuscript.

Line 380-384: *“Even in the absence of a nectar reward, moths showed perseverance by probing longer on the humid flower than on the ambient flower (Fig.4g). Moths also entered the humid flower more frequently while probing (Fig. 4h). Flower entry shows commitment to nectar-feeding, increasing the chances of pollen deposition and export, via dose-dependent pollen loading on moth body parts¹”*

Line 425-427: “Moths benefit by reducing their flower handling cost on humid flowers (Fig. 5a-c), whereas, from the plant perspective, multiple entries of moths into Datura flowers increase pollen loading¹, promote siring success, and cross-pollination^{2,3}.”

It is a bit odd that this desert dwelling plant seems to invest considerable resources (water) in generating this signal, which clearly only works at a very short distance. Is it possible that the humidity is a reward in itself?

We agree and make the point in the revised Introduction and Discussion that the over-the-top investment by the plant is reminiscent of runaway courtship displays in animals, to underscore the substantial metabolic costs (in water balance) invested by these desert plants in maintaining floral humidity. Although the hawkmoths don't spend enough time within/visiting the flowers to derive much of an ambient reward from elevated RH, we discuss the connection between the humidity gradient as a signal and the physiological capacity of the plant to provide copious nectar rewards, with reference to Maynard-Smith and Harper's discussion of “icons” as a class of honest signals in which the provided information is very similar to the actual reward. It is possible however for smaller insects that dwell in floral/cone headspace (eg. Squash bees in squash flowers, yucca moths in yucca flowers, thrips in pollen and ovulate cones of cycads⁴ etc.) to benefit from floral or cone humidity as a rewarding microhabitat.

Line 434-435: “This “microhabitat” hypothesis may also be relevant for the many smaller arthropods that utilize flowers for mating, breeding, and protection without necessarily benefiting the plant⁵”

The risk of dehydration is an issue for most insects, and presumably also for *Manduca*. As such, the humidity would be an honest signal. As for humidity being an ancient signal. The IRs are indeed of old age, but whether the hygrosensory IRs are as well is uncertain. Moreover, it remains to be shown what the precise function in hygrosensation these IRs have. They are necessary for proper hygrosensation but are likely not the actual receptors.

We recognize that the final paragraph is somewhat speculative and difficult to test. However, it is intriguing to consider flowers in the late Jurassic as small habitats for mating or brooding insects that were co-opted into serving as pollinators through responding to local RH as well as CO₂, heat, and scent as inherently interesting sensory stimuli.

The gap in understanding the transduction mechanism in hygrosensation is an open question in the field of sensory physiology/neuroscience as is the transduction mechanism for temperature sensing. Although these questions are ripe for further investigations, they are beyond the scope of this manuscript.

Very minor comment:

Line 242, wrong figure reference...

Thank you, now corrected (line 253)

References

- 1 Smith, G., Kim, C. & Raguso, R. A. Pollen accumulation on hawkmoths varies substantially among moth-pollinated flowers. *bioRxiv*, doi:10.1101/2022.07.15.500245 (2022).

- 2 Elle, E. & Hare, J. D. Environmentally induced variation in floral traits affects the mating system in *Datura wrightii*. *Funct. Ecol.* **16**, 79-88, doi:<https://doi.org/10.1046/j.0269-8463.2001.00599.x> (2002).
- 3 Bronstein, J. L., Huxman, T., Horvath, B., Farabee, M. & Davidowitz, G. Reproductive biology of *Datura wrightii*: the benefits of a herbivorous pollinator. *Ann. Bot.* **103**, 1435-1443, doi:10.1093/aob/mcp053 (2009).
- 4 Terry, L. I., Roemer, R. B., Walter, G. H., Booth, D. & Lee, K. P. Thrips' responses to thermogenic associated signals in a cycad pollination system: the interplay of temperature, light, humidity and cone volatiles. *Funct. Ecol.* **28**, 857-867, doi:10.1111/1365-2435.12239 (2014).
- 5 Cardoso, J. C. F., Gonzaga, M. O., Cavalleri, A., Maruyama, P. K. & Alves-Silva, E. The role of floral structure and biotic factors in determining the occurrence of florivorous thrips in a distylous shrub. *Arthropod-Plant Interactions* **10**, 477-484, doi:10.1007/s11829-016-9443-y (2016).

Response to reviewer #2

My curiosity was piqued by the abstract of this manuscript when I read that it was about relative humidity in the floral headspace being perceived by hawkmoths and furthermore, that hawkmoths preferred more humid headspaces. Until then, I had not known that humidity could be used as such by pollinators. I was therefore surprised upon reading the manuscript that this had been documented before and so my main question was to find out what was novel about this paper. Upon reading further, I found that the main angle was in finding out whether floral humidity is a floral cue or a floral signal. The differences between these (according to this manuscript) is that floral cues usually benefit just one party and typically are not costly to produce, whereas floral signals usually benefit both parties and typically incur some cost to the producer. For example, floral pigments may be regarded as a signal because they increase plant fitness and also improve pollinator foraging efficiency. They are also costly to produce. The authors suggest that recent literature views floral humidity as a cue, although I can see that there are potential arguments for or against whether it benefits pollinators. While the authors argue that floral humidity is more likely a signal than a cue, I am not sure that they really demonstrate this adequately.

A quick note here – we argue for signal function in this particular system, rather than in all plant-pollinator systems. This was clear from our previous title. Because previous studies regarded floral RH as an unavoidable consequence of nectar evaporation that could be utilized by pollinators to track floral profitability (a cue), our current study expands the role of floral RH to signal function. We do not state that floral RH cannot play multiple functional roles in different plant-pollinator systems.

At this stage, I would like to say that the authors did perform some truly beautiful and elegant experiments. They were wonderfully innovative and used a host of interesting, disparate, multi-disciplinary techniques to test their hypotheses. If ever there was a case to publish a paper just for the artfulness of the experiments – this would be it. However, weighing up against the beauty of the experiments is the importance and novelty of the question and also whether the question was properly tested. Unfortunately, I could not get excited about this aspect of the manuscript: I

found the distinction between signals and cues somewhat semantic and niche, but also very poorly developed. As a pollination biologist, I have never really seen the need to distinguish between them and the authors do not provide great reason why we should. I initially thought that this kind of information was lacking because the authors had tried to fit too much information into a small amount of space, but in fact, the manuscript is quite long. There are 5 multi-panel figures (one of them has 13 panels and sub-panels, another has 12). I think there is a general need for the authors to think more carefully about what they want the main angle of the paper is and how best to show/test it, as many of the things done in this paper (although cool) did not link up easily with the question at hand.

First, we would like to thank the reviewer for a thorough review and for recognizing our efforts in addressing a relatively understudied topic in pollination biology. The manuscript has significantly benefitted from these criticisms, and we have attempted to address all of them in our revision.

It is worth noting a few general points here before addressing more specific questions.

First, we are members of a Neurobiology and Behavior department and are interested in a full spectrum of sensory-mediated organismal interactions that include but are not limited to pollination. Thus, one of our guiding perspectives is that traits do not have fixed roles in communication, such that the present study cannot definitively demonstrate that floral humidity is only (ever) a signal vs. a cue, and this was not our goal. Instead, we selected an ideal, well-studied system in which to thoroughly describe the functions and limitations of floral humidity.

Second, the “signal vs. cue” conundrum is a central question in the study of animal communication and is not regarded as semantic or niche among scientists concerned with information theory or animal behavior¹⁻⁵. While pollination is a large field that touches upon many subdisciplines, many important questions in pollination (How do floral traits evolve? Are floral phenotypes integrated? How do pollinators balance innate vs. learned preferences? How flexible is foraging behavior? etc.) are impacted by whether floral traits can be regarded as signals or cues to pollinators and other floral visitors. Testing how natural selection acts upon those traits, while not the purpose of this manuscript, is a major question in our field. We have attempted in this revision to distinguish more clearly between categories of signals and cues, and their respective fitness consequences to sender and receiver, and to explain why that distinction is important, both to this study and in general.

Finally, the reviewer raises a larger philosophical question about novelty and primacy in science, while complimenting the originality and quality of our experiments. The study of floral humidity (including our previous work⁶) remains sufficiently new/rare that there is no definitive experimental study or review establishing the full conceptual scope of this phenomenon. Although our study is focused on one model system, we believe that we have addressed questions from both flower and pollinator perspectives with greater rigor and completeness than in previous studies on this theme.

Perhaps the key point of concern expressed by the reviewer is that the questions (largely ultimate, functional, or fitness-related) framing the study seem ill-matched to (many of) the experimental approaches (often proximate or mechanism-related) taken here. In our revision, we address these concerns by providing new data and additional analyses of our existing data sets, and by drawing stronger connections to the findings of previous studies, including those of our colleagues, on the *Datura-Manduca* model system. We hope that these revisions have clarified the value and novelty of our study.

Reviewer 2: “I was a little frustrated that as I came to the end of the introduction, there was not any sign of an hypothesis. I later found hypotheses embedded in various tables, figure legends and results. I find that having some broad, clear hypotheses in the introduction is important because it prepares the reader for the results, making them easier to interpret.”

We regret this oversight, and we are grateful for the opportunity to revise our text. We have re-framed much of the text and now outline specific hypotheses towards the end of the introduction, to better prepare the reader for the results discussed below:

Line 81-82: *“Given the rapid dissipation of floral RH from the narrow nectar tube and open petals of *O. cespitosa*⁶, we hypothesized that flowers with trumpet-shaped corollas might remain humid longer (also see⁷).”*

Line 97-99: *“By experimentally decoupling floral humidity from nectar presence in artificial flowers, we tested three alternative hypotheses regarding the functional role of floral humidity in this system: neutral trait, honest trait, and unreliable trait.”*

Reviewer 2: “What I found was a manuscript that was not easy to follow. I think this was often because it was seldom clear how the research at hand related to the broad question. Just as an example: why were the authors looking for vertical humidity gradients? How does this relate to the main question (cue or signal)? I think it may be very helpful to structure the headings and subheadings as questions. However, it is also important for the reader to know why each question is important. For example, why is it important to know why wing fanning can influence floral humidity?”

Thank you for this point. We have attempted to improve the readability and clarity of our study by –

1. Modifying the subheadings of the results section to read as conclusions for each section.
2. Leading each subsection with a question that informs the reader about the motivation for performing the specific experiment.

These modifications have been highlighted in the results section.

Line 113: *“At what spatial scale might floral RH be relevant, and for how long after flowers open?”*

Line 135: *“How robust are floral humidity gradients to wind and other disturbances?”*

Line 149-152: *“We hypothesized that floral humidity in *Datura* may result from the accumulation of saturated air in the floral headspace through development from the bud stage and that dissipation of the humidity is prevented by the conical architecture of the flower. Accordingly, we predicted that a moth’s visit to the flower should deplete floral humidity due to the rapid (~25 Hz) wing fanning of a hovering moth.”*

Line 175: *“Do flowers contain stomates on the corolla to facilitate water vapor emission, as leaves do?”*

Line 186: *“How do moths sense floral humidity gradients?”*

Line 198-199: *“Although insects can sense ambient humidity, it is unclear whether the range of floral humidity is sufficient to trigger robust responses by the hygrosensory neurons of pollinators”*

Line 206: *“How can we design an appropriate humidity stimulus?”*

Line 247: *“Do moths show an innate preference for humid flowers?”*

Line 281: *“Do moth preferences change in the presence of nectar?”*

Reviewer 2: *“Hypothesis testing: The main angle of the paper was that the authors try to distinguish between whether relative humidity is a cue or a signal. However, I found that the hypotheses were not clear enough to do this and I also do not think that the tests were appropriate. The 2 alternatives (cue versus signal) were briefly defined as follows: Cues are when only 1 partner benefits and when the trait is not costly to produce, while signals are when both partners benefit and when the trait is costly to produce. In fact, when you break it down, there are also categories which potentially don't count as a cue or a signal (e.g. when there is a cost, but only 1 partner benefits), and it is possible that RH falls into one of those un-named categories (see table 1). While I have tried to visualize the different categories in Table1, this visualization may be quite simplistic because there may be instances when one partner is actively harmed by a signal (but perhaps this counts as no-benefit)”*.

We thank the reviewer for providing a table of possible outcomes, to which we respond here, considering “partner 1” to be the sender of the information (the flower) and “partner 2” to be the recipient of the information (the pollinator). Then, below, we describe the revisions to our text that address the reviewer’s questions.

Cost	Partner 1	Partner 2	Classification
------	-----------	-----------	----------------

Yes	Benefit	Benefit	Signal
-----	---------	---------	--------

(This satisfies the classical definition of a signal)

No	No-benefit	Benefit	Cue
----	------------	---------	-----

(This is a cue. Concerning the Animal Behavior literature, although nearly any trait can be demonstrated to have a cost, traditional cues such as human foot odor or CO2 exhaled in breath, are considered unavoidable consequences of metabolism rather than a phenotypic investment, and can be eavesdropped upon by predators or parasites, e.g. mosquitoes).

No	Benefit	Benefit	?
----	---------	---------	---

(It is hard to think of an example of this in a floral context. Facultative insect visitors residing in flowers may attract pollinators in ways that confer collateral benefits, but this is neither a signal nor a cue, nor strictly speaking, a plant trait).

Yes	No-Benefit	Benefit	?
-----	------------	---------	---

(In the context of signaling theory, such a scenario would result in strong, negative selective pressure against the trait in question. Recent studies of crickets in Hawaii indicate that costly traits that normally confer fitness advantages (loud male songs) are selected against [and become quite rare] when a signal parasite [parasitoid fly] was introduced to Hawaii⁸. One could

imagine extreme cases of florivory swamping the benefits of pollination when an introduced florivore (e.g. strawberry bud weevil) attacks flowers with the brightest visual display. This outcome, while interesting, is not the focus of our study.)

The distinction between signals and cues emerges from extensive research in the field of animal behavior for several decades, including seminal studies by Zahavi⁹, Maynard-Smith³, Dawkins¹⁰ and summarized in major textbooks by Bradbury and Vehrencamp⁵. Signaling theory has entered the realm of pollination biology from the standpoint of pollinator cognition and ethology and remains an active area of research. While it is our responsibility to explain these ideas more clearly in framing our study, they are sufficiently established such that it is not our place to defend whether it is worthwhile to distinguish between cues and signals.

In animal communication, the costs associated with signals are not trivial, homeostatic costs, but rather, the investment in ostentatious songs, plumage and displays, from Tungara frogs to Birds of Paradise. This view is supported by the observation that such displays often draw the attention of predators and cease once mating success has been achieved. Our view is that night-blooming flowers (along with other large, brightly colored and / or strongly scented floral displays) constitute similarly costly displays, which, in parallel, often attract natural enemies and rapidly attenuate (e.g. floral senescence) once flowers have been pollinated.

Given that all previous studies of floral humidity have documented modest ΔRH (e.g. mean ΔRH of 3-5% above ambient humidity levels)^{6,7,11}, our consistent finding of ~10-fold higher ΔRH levels in *Datura* flowers, in spite of challenges to whole-plant and floral water balance in a desert / grassland environment, firmly establish that the night-long maintenance of floral RH in *Datura* bears costs beyond the typical expense of flowering. We detail these costs with new data in the Discussion (see below).

Line 371-374: *“In the case of Datura wrightii, these costs may be especially high owing to the large surface area (164.49 ± 12.28 sq. cm, n=8) and volume (83.95 ± 14.40 cu. cm, n=10) of the flower. Measurements of fresh vs. dry flowers (n=6) show that 87.40 ± 1.54% of flower mass is water, of which fresh nectar mass accounts for only 2.18 ± 0.42%”.*

Concerning the reviewer’s constructive criticisms, we have responded by modifying the main angle of the paper, now presenting a broader narrative that outlines the possible roles of floral humidity, then stressing the role of “honest floral signals” supported by our findings in this system.

Returning to the reviewer’s table of potential cost-benefit outcomes, we have added a contingency table to the revised manuscript to address the most relevant alternative functions for floral humidity (Table 2, see below). In the text, we discuss several possible functional roles floral humidity can play in plant-pollinator interactions and present specific examples for each case. See Line 400-431 in the discussion section.

Table 2. Possible fitness consequences of floral humidity to both plants and pollinators through a cost-benefit contingency table along with broad functional roles. + indicates a fitness benefit, – indicates either a fitness cost or a neutral effect (i.e., no cost). Colors indicate the association of the fitness consequence to specific partners.	
	Pollinator (receiver)

Plant (sender)	-/- (A neutral trait)	-/+ (An exploitable cue)
	+/- (A deceptive signal)	+/ (An honest signal)

Reviewer 2: “Be that as it may, this suggests that there are at least 3 components to distinguishing between these classifications: 1) showing whether partner 1 benefits, 2) showing whether partner 2 benefits, 3) showing whether there is a cost. I question whether the authors do a good job of testing any of these. I could be convinced that the authors test whether RH benefits flowers – here the authors show that humid flowers receive more probes and are visited for a longer duration than less humid flowers. However, whether this has a positive effect on any real measures of fitness (e.g. seed production or pollen transfer) is an open question. In the wild, it is also questionable whether a moth will pass up the opportunity to visit a less humid flower once it gets close enough to assess humidity”...

We appreciate the reviewer’s comments here. One ideal experiment to address this question would require a complete dissociation of floral humidity from living flowers without changing their visual and olfactory display. We have demonstrated that floral humidity in *Datura wrightii* is a consequence of stomatal conductance, which is too dynamic a trait to manipulate without suffering unintended consequences (e.g., other physiological aspects of gas exchange). Less sophisticated experimental manipulations, it is not possible to reduce or remove humidity without damaging the flowers or applying gel on the corolla, which adds undesirable effects to the experiment.

However, one benefit of a well-studied model system is that one’s own work can build upon the discoveries of others. Other research on *Manduca-Datura* interactions includes experiments that directly address pollen export, which reviewer 1 also wished to see, have been performed recently by Smith et al. 2022 (in revision)¹². In this manuscript, the authors show that pollen loading on the proboscis of *Manduca sexta* increases with more visits to *Datura wrightii* flowers. Secondly, we cite Bronstein et al. (2009)¹³ where the authors show that outcrossed pollen from moth visits to *Datura* flowers increases fruit set and the number of seeds per fruit. Thirdly, Johnson et al (2021) show that *Manduca sexta* visits to *Datura wrightii* flowers result in higher seed set than is observed in self-pollinated flowers¹⁴. Finally, Brandenburg et al. (2012)¹⁵ show that petunia flowers (similar and related to *Datura*) that offer less nectar receive fewer instances of probing by *Manduca* moths, ultimately resulting in lower seed set. Taken together, these examples indicate how increased probing and entering of humid flowers by *Manduca* moths would translate to fitness benefits to *Datura* plants.

Reviewer 2: “The next part of the question is admittedly much more difficult to test: is there a benefit to the moth? Here, I cannot see any clear answer, and I do not think that the authors formally test this in any way. It is possible that using RH, moths are able to navigate and manipulate the flowers more easily, hence increasing their foraging efficiency and providing a benefit to the moths. Alternatively, the plants may be exploiting a sensory bias of the moths and tricking them into visiting flowers with little nectar, potentially reducing moth foraging efficiency. This seems like a real possibility since nectar and humidity do not seem to be coupled and moths appear to prefer humid flowers over less humid flowers, even if the less humid flowers have more nectar”...

We appreciate that we had not fully addressed this issue in the original manuscript, but in fact, we have collected data that directly address this question. Here, we have added 3 panels to Fig. 5 that demonstrate the fitness benefit for moths attending to floral humidity. Moths discover nectar sooner when it is paired with a humid flower, whereas they take twice as long to discover nectar when it is paired with an ambient humidity flower. Since the mid-1970s, pollination biologists working on foraging theory¹⁶ have argued that energy intake and profitability resulting from nectar foraging behavior are reasonable surrogates for pollinator fitness (e.g. when such rewards translate into flight capability, territorial defense, use of nectar resources in reproductive fitness, etc.). Here, we calculate the fitness benefit for moths handling humid vs. ambient flowers by showing the net energy expenditure and gain for each hypothesis that we test: neutral trait, honest trait, unreliable trait. Results clearly show that moths benefit from higher foraging efficiency (as a surrogate for fitness) when handling humid flowers (in either neutral or honest trait treatment groups).

The issue of exploiting a sensory bias is complex. The *Datura* system is not deceptive, because nectar rewards generally are rich, and nectar can refill in previously visited plants. Despite the powerful and well documented sensory bias of *Macroglossum*¹⁷ moths and *Bombus* bees¹⁸ for blue flowers, in most cases the blue flowers favored by these insects are rewarding, not deceptive (indeed, Raine and Chittka have argued for local preference for purple flowers by bees in the Alps having resulted from sensory drive). We expand on the possibility for deception in the discussion section-

Line 411-424: *“In Datura wrightii, floral humidity is physiologically decoupled from nectar, presenting potentially conflicting information to its hawkmoth pollinator Manduca sexta. To explore the potential for plant-pollinator conflict in the use of humidity gradients as a signal, we experimentally decoupled the presence of nectar with floral ΔRH (Fig. 5). The response of moths towards the “unreliable trait” manipulation was most interesting. Here, the ambient flower was rewarding but the humid flower was empty, presenting a conflict between the moth’s innate bias and its expectation of a reward. Although we predicted that moths would learn to ignore the empty humid flower, moths visited both flowers equally (Fig. 5f, k, l), seemingly unable to forsake their strong preference for floral humidity. This result presents a potential avenue for plants to use floral RH as a deceptive signal to exploit pollinator perceptual bias, as brood-site deceptive flowers do by mimicking sensory aspects of feces or carrion¹⁹. Datura and Oenothera flowers offer copious nectar to their hawkmoth pollinators, thus, they are not deceptive flowers. However, our findings predict that hawkmoth-pollinated flowers shown to lack nectar (e.g., Plumeria rubra, Brassavola nodosa) would benefit by adding humidity gradients to their visually conspicuous, fragrant flowers^{20,21} (Table 2, deceptive).”*

Reviewer 2: “The third part of the question (is humidity production costly?) is not a simple question either and I would argue that everything on a plant or produced by a plant is costly as it takes ATP to produce”...

Semantically, one could make such an argument, but not all costs are comparable, as we explain above in the analogy to ostentatious songs, plumage and displays in animals. There is a literature on the costs (in water balance) to flower production across many kinds of flowering plants, and our model species has the largest leaves and flowers in the Sonoran Desert, suggesting that floral humidity at the unprecedented scale measured in our study is a non-trivial (almost wastefully consumptive) use of water by a desert plant. We address this comment by providing new data on the percent water content of fresh vs. dry *Datura* flowers, while also

discussing other studies of water balance in flowering plants, as compared with the well-established arguments for realized costs (energetics and predation risk) of sexual display and courtship in animals.

Line 371-379: *“In the case of *Datura wrightii*, these costs may be especially high owing to the large surface area (164.49 ± 12.28 sq. cm, $n=8$) and volume (83.95 ± 14.40 cu. cm, $n=10$) of the flower. Measurements of fresh vs. dry flowers ($n=6$) show that $87.40 \pm 1.54\%$ of flower mass is water, of which fresh nectar mass accounts for only $2.18 \pm 0.42\%$. Such an enormous water budget for flowers, along with nearly constant gas exchange through corolla stomata, predicts that drought-stressed plants should produce fewer flowers. Indeed, *Datura wrightii* plants that are water-stressed produce shorter and fewer flowers that yield fewer viable seeds, at a direct cost to reproductive fitness^{22,23}. These physiological costs, like the energetic demands or predation risks associated with vigorous courtship displays^{24,25}, imply that floral humidity must confer significant benefits in the currency of reproductive success.”*

Reviewer 2: “From this point of view, I do not think that the definitions of signals versus cues have been particularly well thought-out. Perhaps a better question is whether the trait evolved in response to the other partner for communication purposes or whether the trait evolved for a different function (other than communication) and is simply being used by the other partner.

We appreciate this comment, but we have never been comfortable with that argument in the animal behavior literature, because it implies that clear, compelling phylogenetic arguments can be made for both interacting lineages in a coevolutionary context. It is very difficult to discern what a trait evolved for in deep time, by examining the question in present (ecological) time. Our study has focused on interactions between hawkmoths and *Datura wrightii*. Answering the reviewer’s query would require a phylogenetic and physiological study of the entire genus *Datura*.

My guess is that RH in floral tubes may be important because it reduces the evaporation of nectar or prevents flowers from wilting. I doubt that it evolved for pollinator communication. But I may be wrong – the authors have not tested this.

We have already described one example in which floral RH is simply an ephemeral, evaporative consequence of nectar presence, in *Oenothera* flowers. This is clearly a cue, shown to be useful to foraging moths and unavoidable to the plant. The current study presents a very different outlook, which calls attention to trait diversity across floral phenotypes rather than casting doubt on the conclusions of the previous study on *Oenothera*. We would not conclude that floral humidity has only one biological function or evolutionary origin.

But these questions expose the fact that the main angle of this manuscript (distinguishing between signals and cues has not been properly thought through). This is exemplified by the statement in L94: “The question of whether floral humidity may also function as a signal (like floral color and scent) hinges upon whether RH enhances floral fitness”. But this completely ignores the other 2 components of distinguishing signals from cues (see table 1)”...

We address this concern by elaborating on the possible functional roles of floral humidity in the discussion section (Table 2, Line 400-431). In summary, we have shown that floral RH is costly, that foraging moths benefit by responding to it, and that humid flowers are likely to benefit through increased pollinator visitation, pollen removal and siring success.

Reviewer 2: Lastly, contributing to the difficulty in reading this paper, I often found the writing in the manuscript quite awkward. Below, I highlight some examples of this from the introduction:

L42: “The recent expansion of studies in plant-pollinator interactions reflects increased recognition of their importance to ecosystem services, agricultural food security and biological diversity...” I found that this opening paragraph did not provide a strong hook to catch the reader. This paragraph did not make me want to read further and engage with the manuscript, it just felt as though the authors were trying to squash their interesting research into some applied box. It stands isolated and apart from the rest of the manuscript because it is.

We have fundamentally changed the opening paragraph of the introduction. Originally, we felt it would be necessary to frame our study broadly for readers who do not already appreciate the full spectrum of pollination-related questions and fields. Given these comments, we were happy to change the opening paragraph to a more focused introduction of our aims:

Line 43-44: *“The spatial scale at which pollinators are attracted by floral traits has important consequences for pollinator foraging efficiency²⁶, resource partitioning²⁷, and plant gene flow^{28,29}”*

L43: “Examining plant-pollinator interactions through a neuroethological lens presents unique opportunities to investigate the sender-receiver elements of ecological communication.” The authors need to be aware that their use of language is not easily accessible to readers. This sentence, which is full of jargon is a case in point.

We have deleted this sentence

L47: “Plant-pollinator interactions are maintained through floral signals and the innate responses they evoke in the target pollinator.” Is this not over-stressing the role of innate responses? Surely learned responses can also be very powerful. – think mimicry.

We have deleted this sentence.

L57 Shouldn't this be “relative humidity (RH)”

Yes, now corrected.

L80 In a lab essay

Now corrected as *“In the laboratory”*

L81 non-rewarding rather than unrewarded

Corrected to non-rewarding

L82 “pollinators ...can associate both dry and humid flowers with nectar presence under varied levels of background humidity, thus highlighting their flexible valence to floral humidity.” Again, this language is not easily accessible

Modified the sentence to:

*Line 72-73: “Similarly, generalist pollinators like the bumblebee, *Bombus terrestris*, can discriminate ΔRH on artificial flowers when paired with a reward in a lab setting³⁰”*

L87 “Harrap and Rands 35 followed up their garden survey with manipulative experiments on two of their most humid flowers: *Calystegia sylvatica* and *Escholtzia californica*, yielding two important insights. First, funnel shaped morning glory flowers produced the highest Δ RH values (~3.7%) indicating that floral shapes that enclose the headspace can retain higher humidity levels³¹.” The authors are maybe not being critical enough of the work they cite: by comparing the RA of 2 flowers, one can't say definitively that it is floral shape which is influencing RA. It could be any other unmeasured trait.

We have deleted these sentences. Instead, we pose it in the form of a hypothesis that floral architecture may influence floral humidity gradients, even when nectar is not copious.

L94: “The question of whether floral humidity may also function as a signal (like floral color and scent) hinges upon whether RH enhances floral fitness” Here I suddenly feel a little unsure about the distinctions between signals (both benefit, costly) and cues (one benefits, not costly). Surely most floral traits that are attractive to pollinators should benefit both the plant and the pollinator? In this case CO₂ should be regarded as a signal because it benefits both.

This is not necessarily true. Our (admittedly incomplete) understanding of floral CO₂ is that it is an unavoidable by-product of flower bud growth, nectar secretion and scent production in most plants, and is ephemeral (dissipating shortly after anthesis). Our previous work on this trait, in the *Manduca-Datura* system, suggested that CO₂ largely benefits foraging moths (a cue) because it is more temporally correlated with nectar availability than scent or visual display. Pollinator priorities do not always match those of plants, and vice-versa. Conversely, carrion flowers release CO₂ as a consequence of thermogenesis and thereby attract pollinators via deceptive means, often at a fitness cost to the insects. These are very different outcomes.

As should humidity because others have demonstrated that it is attractive to pollinators. You may point out that there is a difference in costliness. So what if it benefits both but is not costly?

This is an interesting semantic point but there is little evidence for mutually beneficial signals that bear no cost, whether in animal communication or in plant-pollinator communication. Third party damages (e.g. florivory and larceny / nectar robbery) are universal, whether due to yeasts, weevils, deer or monkeys, and these outcomes often result from attraction of these third parties (or their vectors) to flower color and/or scent. Costs.

Or if it benefits one but is costly. For example floral scents or colors in mimetic flowers are costly and only benefit the plant. I do not feel that the differences between cues and signals in plants is properly covered or that it is important.

There is a large literature and a recent book by Schiestl and Johnson³¹ addressing these examples, and most authors concur that floral traits in mimetic or deceptive flowers are parasitic on actual fitness-related signals (at least in sexual mimicry and food (nectar, pollen) deceptive flowers) that are associated with other, mutually beneficial interactions. The case of brood site deception is different because the flower is mimicking cues from dead or inanimate resources (carrion, feces, rotting fungi). This latter case is beyond the scope of our study.

Or that this manuscript actually helps us to distinguish whether humidity is a signal or a cue (mostly because RH does not fit properly into either of the two definitions given”).

We have addressed this issue both in the introduction and discussion sections of the revised manuscript.

See line 83 to 88: “Furthermore, if above-ambient floral RH persists after nectar has been extracted by an earlier visitor, the disconnect between floral humidity and nectar status may present conflicting information to subsequent floral visitors. If floral humidity and nectar are physiologically decoupled, this may expand the possible roles of floral RH from a profitability cue for pollinators to a floral signal, such as color or scent. The potential signal function of RH, whether deceptive or honest, hinges upon whether it enhances the fitness of one or both partners in plant-pollinator communication^{32,33}”

L102: why would evaporation make RA (RH?) an ephemeral cue? Nectar is often constantly replenished. I am also unsure that nectar removal would immediately make floral tubes suddenly not-humid. I expect that humidity would remain for a long time after nectar removal (even if it was the nectar that produced the humidity in the first place).

These comments are addressed by our results on efficacy (in wind, in presence of hovering moths), combined with the data on stomatal blockage (Fig. 1 in manuscript). Our primary finding is that vigorous stomatal conductance replenishes floral RH quickly and independent of nectar presence. Without this active process, humidity in a floral tube would dissipate into drier ambient air, just as CO₂ does, as was shown in our previous studies.

Nectar secretion patterns vary enormously among flowering plants, hence the study of “bonanza-blank” species³⁴, steady state and “last dregs”^{35,36} examples during the height of the “optimal foraging” era. Specifically, nectar in night blooming plants varies greatly in standing crop and secretion rate.

In the supplementary figure 1c we have added the contribution of standing nectar or water to the floral humidity of a flower. To do so we made an artificial flower using a funnel and falcon tube (Supplementary figure 1d) that matches the geometry of *Datura* flowers. We added 200ul of *Datura* nectar (the max standing crop volume offered by flowers in greenhouse plants) or water and measured the vertical humidity gradient. Our data show that even with a generous volume of nectar or water, floral humidity as an outcome of fluid evaporation is typical of studies with other plants but is trivial in comparison to what *Datura* flowers produce through stomatal gas exchange.

Regarding the reviewer’s specific comment about nectar depletion and floral humidity, these concerns were addressed directly in the findings of Von Arx et al (2012)⁶, a former study from our lab, in which we performed manipulations on *Oenothera cespitosa* flowers and mapped floral RH with and without nectar. We found that nectar removal significantly decreased the substantially lower Δ RH of *O. cespitosa* flowers, which is not the case with *Datura wrightii*.

L107: “Furthermore, their attraction towards humid flowers cannot be abolished by experimentally rewarding only the ambient flowers, indicating a receiver-bias towards floral RH.” While I understand what the authors are trying to say in this sentence, I find the wording very awkward.

We have removed this sentence from the current version.

References

- 1 Laidre, M. E. & Johnstone, R. A. Animal signals. *Curr. Biol.* **23**, R829-R833, doi:10.1016/j.cub.2013.07.070 (2013).

- 2 Lichtenberg, E. M., Heiling, J. M., Bronstein, J. L. & Barker, J. L. Noisy communities and signal
detection: why do foragers visit rewardless flowers? *Philosophical Transactions of the Royal
Society B: Biological Sciences* **375**, 20190486, doi:doi:10.1098/rstb.2019.0486 (2020).
- 3 Smith, M. J. & Harper, D. G. C. Animal signals: Models and terminology. *J. Theor. Biol.* **177**, 305-
311, doi:<https://doi.org/10.1006/jtbi.1995.0248> (1995).
- 4 Essenberg, C. J. Intraspecific relationships between floral signals and rewards with implications
for plant fitness. *AoB PLANTS* **13**, doi:10.1093/aobpla/plab006 (2021).
- 5 Bradbury, J. W. & Vehrencamp, S. L. *Principles of animal communication, 2nd ed.* (Sinauer
Associates, 2011).
- 6 von Arx, M., Goyret, J., Davidowitz, G. & Raguso, R. A. Floral humidity as a reliable sensory cue
for profitability assessment by nectar-foraging hawkmoths. *Proc Natl Acad Sci U S A* **109**, 9471-
9476, doi:10.1073/pnas.1121624109 (2012).
- 7 Harrap, M. J. M., Hempel de Ibarra, N., Knowles, H. D., Whitney, H. M. & Rands, S. A. Floral
humidity in flowering plants: A preliminary survey. *Frontiers in Plant Science* **11**,
doi:10.3389/fpls.2020.00249 (2020).
- 8 Zuk, M., Rotenberry, J. T. & Tinghitella, R. M. Silent night: adaptive disappearance of a sexual
signal in a parasitized population of field crickets. *Biol. Lett.* **2**, 521-524,
doi:10.1098/rsbl.2006.0539 (2006).
- 9 Zahavi, A. & Zahavi, A. *The handicap principle: A missing piece of darwin's puzzle.* (Oxford
University Press, 1999).
- 10 Guilford, T. & Dawkins, M. S. Receiver psychology and the evolution of animal signals. *Anim.
Behav.* **42**, 1-14, doi:[https://doi.org/10.1016/S0003-3472\(05\)80600-1](https://doi.org/10.1016/S0003-3472(05)80600-1) (1991).
- 11 Harrap, M. J. M. & Rands, S. A. The role of petal transpiration in floral humidity generation.
Planta **255**, 78, doi:10.1007/s00425-022-03864-9 (2022).
- 12 Smith, G., Kim, C. & Raguso, R. A. Pollen accumulation on hawkmoths varies substantially among
moth-pollinated flowers. *bioRxiv*, doi:10.1101/2022.07.15.500245 (2022).
- 13 Bronstein, J. L., Huxman, T., Horvath, B., Farabee, M. & Davidowitz, G. Reproductive biology of
Datura wrightii: the benefits of a herbivorous pollinator. *Ann. Bot.* **103**, 1435-1443,
doi:10.1093/aob/mcp053 (2009).
- 14 Johnson, C. A. *et al.* Coevolutionary transitions from antagonism to mutualism explained by the
Co-Opted Antagonist Hypothesis. *Nature Communications* **12**, doi:10.1038/s41467-021-23177-x
(2021).
- 15 Brandenburg, A., Kuhlemeier, C. & Bshary, R. Hawkmoth pollinators decrease seed set of a low-
nectar *Petunia axillaris* line through reduced probing time. *Curr. Biol.* **22**, 1635-1639,
doi:<https://doi.org/10.1016/j.cub.2012.06.058> (2012).
- 16 Pyke, G. H. Plant-pollinator co-evolution: It's time to reconnect with Optimal Foraging Theory
and Evolutionarily Stable Strategies. *Perspect. Plant Ecol. Evol. Syst.* **19**, 70-76,
doi:<https://doi.org/10.1016/j.ppees.2016.02.004> (2016).
- 17 Kelber, A. Innate preferences for flower features in the hawkmoth *Macroglossum stellatarum*. *J.
Exp. Biol.* **200**, 827-836 (1997).
- 18 Raine, N. E. & Chittka, L. The adaptive significance of sensory bias in a foraging context: Floral
colour preferences in the bumblebee *Bombus terrestris*. *PLOS ONE* **2**, e556,
doi:10.1371/journal.pone.0000556 (2007).
- 19 Jürgens, A., Wee, S. L., Shuttleworth, A. & Johnson, S. D. Chemical mimicry of insect oviposition
sites: a global analysis of convergence in angiosperms. *Ecol. Lett.* **16**, 1157-1167,
doi:10.1111/ele.12152 (2013).
- 20 Schemske, D. W. Evolution of floral display in the orchid *Brassavola nodosa*. *Evolution* **34**, 489-
493, doi:10.2307/2408218 (1980).

- 21 Haber, W. A. Pollination by deceit in a mass-flowering tropical tree *Plumeria rubra* L. (apocynaceae). *Biotropica* **16**, 269-275, doi:10.2307/2387935 (1984).
- 22 Elle, E., van Dam, N. M. & Hare, J. D. Cost of glandular trichomes, a "resistance" character in *Datura wrightii* regel (solanaceae). *Evolution* **53**, 22-35, doi:10.1111/j.1558-5646.1999.tb05330.x (1999).
- 23 Elle, E. & Hare, J. D. Environmentally induced variation in floral traits affects the mating system in *Datura wrightii*. *Funct. Ecol.* **16**, 79-88, doi:<https://doi.org/10.1046/j.0269-8463.2001.00599.x> (2002).
- 24 Marler, C. A. & Ryan, M. J. Energetic constraints and steroid hormone correlates of male calling behaviour in the túngara frog. *J. Zool.* **240**, 397-409, doi:<https://doi.org/10.1111/j.1469-7998.1996.tb05294.x> (1996).
- 25 Bernal, X. E., Rand, A. S. & Ryan, M. J. Acoustic preferences and localization performance of blood-sucking flies (*Corethrella* Coquillett) to túngara frog calls. *Behav. Ecol.* **17**, 709-715, doi:10.1093/beheco/arl003 (2006).
- 26 Kulahci, I. G., Dornhaus, A. & Papaj, D. R. Multimodal signals enhance decision making in foraging bumble-bees. *Proc Biol Sci* **275**, 797-802, doi:10.1098/rspb.2007.1176 (2008).
- 27 Goldshtein, A. *et al.* Reinforcement learning enables resource partitioning in foraging bats. *Curr. Biol.* **30**, 4096-4102.e4096, doi:<https://doi.org/10.1016/j.cub.2020.07.079> (2020).
- 28 Skogen, K. A., Overson, R. P., Hilpman, E. T. & Fant, J. B. Hawkmoth pollination facilitates long-distance pollen dispersal and reduces isolation across a gradient of land-use change. *Annals of the Missouri Botanical Garden* **104**, 495-511, 417 (2019).
- 29 Deng, J.-Y., van Noort, S., Compton, S. G., Chen, Y. & Greeff, J. M. Conservation implications of fine scale population genetic structure of *Ficus* species in South African forests. *For. Ecol. Manage.* **474**, 118387, doi:<https://doi.org/10.1016/j.foreco.2020.118387> (2020).
- 30 Harrap, M. J. M., Hempel de Ibarra, N., Knowles, H. D., Whitney, H. M. & Rands, S. A. Bumblebees can detect floral humidity. *J. Exp. Biol.* **224**, doi:10.1242/jeb.240861 (2021).
- 31 Johnson, S. D. & Schiestl, F. P. *Floral Mimicry*. (Oxford University Press, 2016).
- 32 Wright, G. A. & Schiestl, F. P. The evolution of floral scent: the influence of olfactory learning by insect pollinators on the honest signalling of floral rewards. *Funct. Ecol.* **23**, 841-851, doi:<https://doi.org/10.1111/j.1365-2435.2009.01627.x> (2009).
- 33 Schiestl, F. P. & Johnson, S. D. Pollinator-mediated evolution of floral signals. *Trends Ecol. Evol.* **28**, 307-315, doi:<https://doi.org/10.1016/j.tree.2013.01.019> (2013).
- 34 Brink, D. A Bonanza-Blank Pollinator Reward Schedule in *Delphinium nelsonii* (Ranunculaceae). *Oecologia* **52**, 292-294 (1982).
- 35 Whitham, T. G. Coevolution of foraging in *Bombus* and nectar dispensing in *Chilopsis*: A last dreg theory. *Science* **197**, 593-596, doi:doi:10.1126/science.197.4303.593 (1977).
- 36 Hodges, C. M. & Wolf, L. L. Optimal foraging in bumblebees: Why is nectar left behind in flowers? *Behav. Ecol. Sociobiol.* **9**, 41-44, doi:10.1007/BF00299851 (1981).

Reviewers' Comments:

Reviewer #1:

Remarks to the Author:

My minor-ish concerns have all been addressed.

Reviewer #2:

Remarks to the Author:

I like the paper a lot, especially for the multidisciplinary approach taken. I think that the authors have gone to great lengths to improve this manuscript and it reads a lot better than the previous version. In particular, I found that the new headings made the manuscript a lot clearer and the hypotheses more accessible. I only have one major criticism of the manuscript and that is the classification of RH as an honest signal. I know that the authors went to great effort to try and show this experimentally, but you will see from my comments that I am still not at all convinced by this aspect of the paper. I don't necessarily think that it is that important to show whether the trait is honest or not for the acceptance of the manuscript, but I do think it is important not to over-interpret the data.

ABSTRACT: I liked this new abstract

INTRODUCTION:

I have often observed pollinators (bees and butterflies) approaching flowers, apparently inspecting them from very close quarters (without landing) before probing or more often passing the flower by. It is unclear what is governing these choices once attraction has already occurred, but it could be possible that RH is playing a role. It may be worth adding references to such behaviour in the first paragraph of the introduction if you are aware of any published accounts.

I enjoyed reading the intro this time round and especially liked the last paragraph, where the significance of this work is made much clearer. There are several instances here and throughout the ms where genera (e.g. *Manduca* and *Datura*) are not italicised.

RESULTS

The new subheadings are extremely helpful

L19: what is an "index cue"?

L21: Not clear what the role was expanded from

L27: The word "abolished" seems out of place. Perhaps: Moths can track and are attracted to minute changes in RH, perceived by antennal hygrosensory sensilla. However they are not attracted to humid flowers when these sensilla are experimentally occluded.

L54: Index cues?

L82: longer than what? I don't recall this being a comparative study of RH in different tube phenotypes

Fig 1A: I was not able to see the grey shading referred to in the legend.

Fig 1E: The last line of the legend (for 1F) is useful as it is a very brief explanation of what we are seeing and why it is important. Something similar for the other panels may be helpful. E.g. for 1E: Letters show pairwise comparisons using the Wilcoxon test with Bonferroni correction, suggesting that the unusually high floral humidity in *Datura* is not an artifact of greenhouse conditions

L156: this is remarkable

L195: Not sure what you mean by fictive (used here and elsewhere in the text and figures). Dictionary

definition is that it is imaginative (similar to fiction writing)

Table 1. I really like the idea of the table but have some trouble with the determination of trait honesty and feel that it is too simplistic. In this system, humidity is decoupled from reward so that humid flowers can sometimes provide a reward and sometimes not. In other words, ambient flowers and humid flowers should both have plus and minus signs in your table, not just a single sign. To me, this is not an honest signal and it appears as though individuals could use it to exploit pollinator sensory bias: get more visits even though there is no nectar in the flower. Essentially, I am evaluating honesty based on whether individuals (which are arguably the currency of natural selection) are being honest or not. However, it appears that the authors are evaluating honesty based on population averages, which feels groups selectionist to me. It may well be useful in explaining how we expect moths to respond to these traits, but it does not tell us about how the traits are likely to evolve and invade populations. I think that if you were able to show that in the wild, like *Oenothera*, there is a correlation between nectar volume and RH, then there would be argument to say that it is an honest trait. And you may well be right – that in many systems this is an honest trait. However, this system seems somewhat unique in that RH is a super-stimulus and is much higher than what is found in most other systems. It strikes me as being similar to the enormous displays of many deceptive orchids. There is a distinct possibility that this RH stimulus has evolved to exploit the sensory bias of moths evolving in response to other plants which do use RH as an honest signal (e.g. *Oenothera*).

If I can draw a parallel between nectar production and flower color: nectar quality in many species varies within populations and can be heritable, so that some individuals produce lots of nectar while others produce little or none. In these systems, flower color is used to signal reward, and in most cases, color will be associated with a reward. But many flowers have no rewards (either because they have been visited or because they inherently do not have nectar). But they still signal to insects to come and visit. In this case, color can not be regarded as an honest signal because it allows cheating within an otherwise honest system. In contrast, some plants have color signals that change upon visitation, a signal to pollinators, indicating a lack of nectar. I would regard this as an honest signal, because it does not allow any cheating within the system.

Similarly, I would regard RH as an honest signal if it changed in relation to nectar volume. You show that RH signals are restored within seconds of nectar removal so that nectarless flowers are still emitting a strong RH signal, suggesting dishonesty. An honest signal would be if the RH signal restored itself at the same rate as the nectar restoration.

All that to say that while it may be worth discussing in an objective way, I do not think that the honesty angle is well supported -at least when thinking about the evolution of such traits and individual fitness. I note that it is discussed later and that the authors take an honesty stance which I think is highly doubtful and takes away from the other great strengths of the manuscript. I think that this section needs some more careful thought. Perhaps the concept of honesty differs depending on whether you think of it in terms of individuals or populations.

L287: "Across all experimental groups, moths found nectar sooner when humid flowers were paired with nectar rewards than when humid flowers were empty" This sentence makes no sense to me. Of course moths will find nectar sooner in flowers with nectar because they should not find any nectar in flowers that are empty. Handling times in rewardless vs rewarding flowers often differ simply because pollinators quickly consume rewards and go versus spend time looking for rewards which are not there.

L424-431: This argument for honest signalling based on evolutionary stability is a weak one because it must be remembered that there are many species of moth pollinated flower. Dishonesty can easily persist because moths have an innate preference for strong RH gradients due to the fact that these gradients are associated with nectar in co-occurring species

Reviewer #1 (Remarks to the Author):

My minor-ish concerns have all been addressed.

Thank You! We appreciate the feedback in this process.

Reviewer #2 (Remarks to the Author):

I like the paper a lot, especially for the multidisciplinary approach taken. I think that the authors have gone to great lengths to improve this manuscript and it reads a lot better than the previous version. In particular, I found that the new headings made the manuscript a lot clearer and the hypotheses more accessible.

Thank you. We appreciate your comments and we have worked hard to address them.

I only have one major criticism of the manuscript and that is the classification of RH as an honest signal. I know that the authors went to great effort to try and show this experimentally, but you will see from my comments that I am still not at all convinced by this aspect of the paper. I don't necessarily think that it is that important to show whether the trait is honest or not for the acceptance of the manuscript, but I do think it is important not to over-interpret the data.

We have now removed the terms "index" and "honest" from the manuscript, in keeping with the reviewer's request that we not over-interpret our results. Instead, we now refer to the second (mutually beneficial) hypothesis outlined at the end of the Intro and in Table 1 as a "reliable trait", rather than an honest one. Our rationale is that we consider signal reliability to be more of a statistical outcome of receiver sampling in sensory markets, whether that be female animals attending a lek of dancing male birds or pollinators visiting (and evaluating) hundreds of flowers within a meadow's "floral marketplace". Although we disagree with the reviewer's strict definition of signal honesty in flowers and can point to arguments made in recent empirical and theoretical studies, we feel that our paper does not suffer by omitting 'honest signal' terminology.

With these changes in the manuscript, we feel that our original title more effectively communicates the message of our paper ("Floral humidity as a signal--not a cue--in a nocturnal pollination system"). Again, we are not claiming here (or anywhere) that floral humidity is always a signal, but we feel that the evidence presented in the present study supports this conclusion for flowers of *Datura wrightii*. If there are no strong objections, we would prefer to revert to the previous title.

ABSTRACT: I liked this new abstract

INTRODUCTION:

I have often observed pollinators (bees and butterflies) approaching flowers, apparently inspecting them from very close quarters (without landing) before probing or more often passing the flower by. It is unclear what is governing these choices once attraction has already occurred, but it could be possible that RH is playing a role. It may be worth adding references to such behaviour in the first paragraph of the introduction if you are aware of any published accounts.

Yes! we share this observation with the reviewer and know that similar observations are commonly made by pollination biologists. We now include a sentence to address that with citations to the following studies:

Corbet, S.A., Kerslake, C.J.C., Brown, D., and Morland, N.E. (1984). Can Bees Select Nectar-Rich Flowers in a Patch? *J. Apic. Res.* 23, 234-242.

Policha, T., Davis, A., Barnadas, M., Dentinger, B.T.M., Raguso, R.A., and Roy, B.A. (2016). Disentangling visual and olfactory signals in mushroom-mimicking *Dracula* orchids using realistic three-dimensional printed flowers. *New Phytol.* 210, 1058-1071.

Line 47-49: "Recently foraged flowers can remain scented, turgid, and pigmented minutes to hours after nectar or pollen has been removed by an earlier visitor, yet it is commonly observed that pollinators reject some flowers upon inspection, without landing" ^{1,2}

I enjoyed reading the intro this time round and especially liked the last paragraph, where the significance of this work is made much clearer. There are several instances here and throughout the ms where genera (e.g. *Manduca* and *Datura*) are not italicised.

Thank you! Now fixed.

RESULTS

The new subheadings are extremely helpful

L19: what is an "index cue"?

We have omitted the word "index" from the manuscript.

An index in animal communication is when the information is physically connected with the object such that it indicates the quality of the sender (For example the roars of red deer stags are constrained by their body size and impact their reproductive success^{3,4}).

L21: Not clear what the role was expanded from

We begin the abstract by stating that previously floral humidity was considered a cue (meaning only one partner can benefit) and that the evidence presented here expands its role to that of a signal (meaning both partners can benefit). We hope that it is clearer now.

L27: The word "abolished" seems out of place. Perhaps: Moths can track and are attracted to minute changes in RH, perceived by antennal hygro-sensory sensilla. However, they are not attracted to humid flowers when these sensilla are experimentally occluded.

The word abolish is common in neurobiology and genetics when referring to the loss of function of genes such as a knockout which abolishes an animal's preference or its innate behavior. We used the word keeping in mind that Nature communications has a broad readership.

We have modified the sentence as –

"Moths can track minute changes in RH via antennal hygro-sensory sensilla, but they lose their attraction to humid flowers when these sensilla are experimentally occluded."

L54: Index cues?

We have omitted the word "index"

L82: longer than what? I don't recall this being a comparative study of RH in different tube phenotypes

We have modified the sentence as-

*“Given the rapid dissipation (~30 mins) of floral RH from the narrow nectar tube and open petals of *O. cespitosa*⁵, we hypothesized that flowers with trumpet-shaped corollas might sustain humidity gradients beyond anthesis”.*

Fig 1A: I was not able to see the grey shading referred to in the legend.

Now modified.

Fig 1E: The last line of the legend (for 1F) is useful as it is a very brief explanation of what we are seeing and why it is important. Something similar for the other panels may be helpful. E.g. for 1E: Letters show pairwise comparisons using the Wilcoxon test with Bonferroni correction, suggesting that the unusually high floral humidity in *Datura* is not an artifact of greenhouse conditions

Thank you! We have now added a sentence to each panel in Fig. 1 and highlighted them in the manuscript.

L156: this is remarkable

We agree – it was quite unexpected

L195: Not sure what you mean by fictive (used here and elsewhere in the text and figures). Dictionary definition is that it is imaginative (similar to fiction writing)

In neuroethological experiments, a classical approach is to capture the essence of the animal's movement and experiences in its natural environment while simultaneously recording the activity of its nervous system. The word “fictive” here means that while moths are head fixed in our electrophysiology setup, they experience an experimental stimulus that matches with their experience of probing *Datura* flowers. “Fictive stimulus” or “fictive locomotion” has been used previously in many published manuscripts and is a common term in the field of sensory neurobiology. The recent explosion in implementing virtual reality setups in animal behavior experiments is another example of the use of fictive stimuli to elicit naturalistic behaviors.

Koning HK, Ahemaiti A, Boije H. A deep-dive into fictive locomotion - a strategy to probe cellular activity during speed transitions in fictively swimming zebrafish larvae. *Biol Open*. 2022 Mar 15;11(3):bio059167. doi: 10.1242/bio.059167. Epub 2022 Mar 22. PMID: 35188534; PMCID: PMC8966775.

Ahrens, M., Huang, K.-H., Narayan, S., Mensh, B., and Engert, F. (2013). Two-photon calcium imaging during fictive navigation in virtual environments. *Frontiers in Neural Circuits* 7.

That said, to make Fig. 2 clearer we have now added an illustration to Fig. 2f, which is the generated stimulus, and corrected the labeling for Fig. 2j which is the fictive stimulus. In addition, we have added a definition sentence to lines 206-208.

“How can we design a fictive humidity stimulus for moths? In neuroethology, a fictive stimulus captures the essence of a sensory stimulus experienced by an animal in its natural environment and presents it in a controlled laboratory experiment⁶”

Table 1. I really like the idea of the table but have some trouble with the determination of trait honesty and feel that it is too simplistic. In this system, humidity is decoupled from reward so that humid flowers can sometimes provide a reward and sometimes not. In other words, ambient flowers and humid flowers should both have plus and minus signs in your table, not just a single sign. To me, this is not an honest signal and it appears as though individuals could use it to exploit pollinator sensory bias: get more visits even though there is no nectar in the flower. Essentially, I am evaluating honesty based on whether individuals (which are arguably the currency of natural selection) are being honest or not. However, it appears that the authors are evaluating honesty based on population averages, which feels groups selectionist to me.

Floral humidity is not inherently decoupled from nectar in this system. *Datura* flowers present copious nectar even before they are completely open. Yes, it can be decoupled if an earlier visitor consumes all the available nectar, but this will also be true for other (generally) honest signals like flower scent and color. Therefore, we are not comfortable with changing the sign to plus minus for a mutually beneficial signal because we have shown evidence for it.

Communication between plants and animals is complex. Flowers rarely present accurate information 100% of the time (maybe only with colored nectar?) which makes it difficult for pollinators to assess honesty based on individual flowers. In our experience and that of many others, the responses of pollinators to empty flowers are not as aversive or salient as are the responses to a punishment (salt, electric shock, quinine, crab spider attack), as shown by Ings and Chittka⁷. Thus, there is often a mismatch between signals and rewards, and pollinators cope with empty flowers in otherwise profitable patches every day.

Individual pollinators are quick at making associations, and if the experience of pollinators with flowers at the population level is positive “on average” then it renders honesty because it enables pollinators to remember specific traits as rewards through reinforcement learning⁸. The signal honesty hypothesis has been tested previously for floral scent^{8,9}, color^{10,11}, and corolla shape¹². The honesty of flower color especially has been tested at a population level in the cited example (local preference for purple flowers associated with on average higher profitability) by Raine and Chittka⁹.

It may well be useful in explaining how we expect moths to respond to these traits, but it does not tell us about how the traits are likely to evolve and invade populations. I think that if you were able to show that in the wild, like *Oenothera*, there is a correlation between nectar volume and RH, then there would be argument to say that it is an honest trait. And you may well be right – that in many systems this is an honest trait. However, this system seems somewhat unique in that RH is a super-stimulus and is much higher than what is found in most other systems. It strikes me as being similar to the enormous displays of many deceptive orchids. There is a distinct possibility that this RH stimulus has evolved to exploit the sensory bias of moths evolving in response to other plants which do use RH as an honest signal (e.g. *Oenothera*).

Deceptive orchids or carrion-mimicking flowers are examples of unreliable signals for pollinators where partners do not share a common interest. These are classic examples where preexisting biases are exploited by plants that offer no benefit to the pollinator such as pollen or nectar rewards. In contrast, the *Datura-Manduca* interaction is an example of mutualism, as both partners benefit from the interaction. This communication system may include avenues for “incomplete honesty” but that is not uncommon in any communication system¹³. As Dawkins and Guilford¹⁴ mention in their benchmark paper about signals, a certain degree of unreliability

will always be present owing to inherent imprecision in phenotype formation, such as developmental noise, and underlying imperfections in the efficacy of the signals.

Certainly, it is possible that *Datura* gains additional visits (and pollen export, etc.) when flowers are empty by exploiting *Manduca* sensory bias for humidity, as suggested by our experimental data. However, as shown in previous work by our group and our colleagues (Alarcon, Bronstein, etc.¹⁵), *Datura* flowers are so richly rewarding, and *Manduca* moths utilize them so heavily for nectar, that overall, their relationship is strongly mutualistic and not deceptive.

If I can draw a parallel between nectar production and flower color: nectar quality in many species varies within populations and can be heritable, so that some individuals produce lots of nectar while others produce little or none. In these systems, flower color is used to signal reward, and in most cases, the color will be associated with a reward. But many flowers have no rewards (either because they have been visited or because they inherently do not have nectar). But they still signal to insects to come and visit. In this case, color can not be regarded as an honest signal because it allows cheating within an otherwise honest system. In contrast, some plants have color signals that change upon visitation, a signal to pollinators, indicating a lack of nectar. I would regard this as an honest signal, because it does not allow any cheating within the system. Similarly, I would regard RH as an honest signal if it changed in relation to nectar volume. You show that RH signals are restored within seconds of nectar removal so that nectarless flowers are still emitting a strong RH signal, suggesting dishonesty. An honest signal would be if the RH signal restored itself at the same rate as the nectar restoration. All that to say that while it may be worth discussing in an objective way, I do not think that the honesty angle is well supported -at least when thinking about the evolution of such traits and individual fitness. I note that it is discussed later and that the authors take an honesty stance which I think is highly doubtful and takes away from the other great strengths of the manuscript. I think that this section needs some more careful thought. Perhaps the concept of honesty differs depending on whether you think of it in terms of individuals or populations.

We have removed the term “honesty” from the manuscript. However, we would like to respond to the reviewer below.

We find that the reviewer’s definition of signal honesty is too stringent. According to their definition, very few signals in insect-plant interaction would qualify as honest. Furthermore, a truly 100% honest signal (perhaps colored nectar in openly accessible flowers in Mauritius¹⁶, where pollen limitation is extreme?) would forfeit the opportunity for a flowering plant to pursue its own fitness optima, as plants and pollinators often show conflicts of interest (many reviews by J.L. Bronstein).

The physical and physiological constraints between nectar secretion and transpiration (humidity) are vastly different. This is also true for other generally honest signals like the scent and color of a flower. Even for the honest signal example cited by the reviewer about flower color change after pollination, we expect the color to change (e.g., in *Lantana camara*) over hours/days after pollination, limited by the dynamics of a post-pollination ethylene response. Would it be a dishonest flower for the time until its color has completely transformed to signal the lack of nectar?

We appreciate the reviewer’s comment, but we are not in the position to challenge an entire field where signal honesty is based on statistical associations across populations of

resources^{4,13,14,17-20}. There is always some variance between floral signals and floral rewards²¹. Our study demonstrates, to our own initial surprise, that this can be true for floral RH in *Datura*.

L287: “Across all experimental groups, moths found nectar sooner when humid flowers were paired with nectar rewards than when humid flowers were empty” This sentence makes no sense to me. Of course moths will find nectar sooner in flowers with nectar because they should not find any nectar in flowers that are empty. Handling times in rewardless vs rewarding flowers often differ simply because pollinators quickly consume rewards and go versus spend time looking for rewards which are not there.

Thank you for this comment. We have modified the sentence as follows to make it clearer

Line 289-290 “Across all experimental groups, the time to reach the nectary was shorter when the humid flower was paired with nectar reward than when the ambient flower was paired with rewards”

L424-431: This argument for honest signalling based on evolutionary stability is a weak one because it must be remembered that there are many species of moth pollinated flower. Dishonesty can easily persist because moths have an innate preference for string RH gradients due to the fact that these gradients are associated with nectar in co-occurring species

This is now lines 427-434 in the current version. We appreciate the reviewer’s concerns about the targets of selection and invasibility, but we want to point out that super-normal floral humidity has not arisen as a novel trait in *Datura wrightii*, but instead may simply reflect the scaling up (of all traits – visual display, scent, and nectar standing crop) in one of North America’s largest flowers, in a genus (*Datura*) of similarly shaped ‘trumpet’ flowers. As we discuss in the text, *D. wrightii* flowers are highly and innately preferred nectar sources for *M. sexta* moths in the Sonoran Desert, as shown by many previous studies¹⁵, and they are also highly and crucially nectar-rewarding plants for both male and female moths. In fact, usage of those other night blooming species (*Mirabilis*, *Oenothera*, *Aquilegia*) as nectar sources for *Manduca* moths is the exception rather than the rule in our Sonoran Desert field sites, as visitation to flowers of *Datura* and (bat pollinated but highly profitable) *Agave* completely dominates the published data sets.

We agree that the flowers can benefit (rather than exploit) from what appears to be a strong sensory bias for floral RH in *M. sexta*, just as previous studies showed strong innate sensory bias for their floral scent²². The fact that on average, *Datura* flowers are highly rewarding, refill nectar, and are highly preferred nectar sources for *M. sexta* moths supports our conclusion that RH is, on average, a reliable and mutually beneficial trait indicating profitability in *Datura* flowers.

References

- 1 Corbet, S. A., Kerslake, C. J. C., Brown, D. & Morland, N. E. Can Bees Select Nectar-Rich Flowers in a Patch? *J. Apic. Res.* **23**, 234-242, doi:10.1080/00218839.1984.11100638 (1984).
- 2 Policha, T. *et al.* Disentangling visual and olfactory signals in mushroom-mimicking *Dracula* orchids using realistic three-dimensional printed flowers. *New Phytol.* **210**, 1058-1071, doi:10.1111/nph.13855 (2016).
- 3 Reby, D. & McComb, K. Anatomical constraints generate honesty: acoustic cues to age and weight in the roars of red deer stags. *Anim. Behav.* **65**, 519-530, doi:<https://doi.org/10.1006/anbe.2003.2078> (2003).

- 4 Laidre, M. E. & Johnstone, R. A. Animal signals. *Curr. Biol.* **23**, R829-R833, doi:10.1016/j.cub.2013.07.070 (2013).
- 5 von Arx, M., Goyret, J., Davidowitz, G. & Raguso, R. A. Floral humidity as a reliable sensory cue for profitability assessment by nectar-foraging hawkmoths. *Proc Natl Acad Sci U S A* **109**, 9471-9476, doi:10.1073/pnas.1121624109 (2012).
- 6 Ahrens, M., Huang, K.-H., Narayan, S., Mensh, B. & Engert, F. Two-photon calcium imaging during fictive navigation in virtual environments. *Frontiers in Neural Circuits* **7**, doi:10.3389/fncir.2013.00104 (2013).
- 7 Ings, T. C. & Chittka, L. Predator crypsis enhances behaviourally mediated indirect effects on plants by altering bumblebee foraging preferences. *Proc Biol Sci* **276**, 2031-2036, doi:10.1098/rspb.2008.1748 (2009).
- 8 Knauer, A. C. & Schiestl, F. P. Bees use honest floral signals as indicators of reward when visiting flowers. *Ecol. Lett.* **18**, 135-143, doi:10.1111/ele.12386 (2015).
- 9 Knauer, A. C., Kokko, H. & Schiestl, F. P. Pollinator behaviour and resource limitation maintain honest floral signalling. *Funct. Ecol.* **35**, 2536-2549, doi:10.1111/1365-2435.13905 (2021).
- 10 Raine, N. & Chittka, L. Colour preferences in relation to the foraging performance and fitness of the bumblebee *Bombus Terrestris Uludag Bee Journal* **5**, 145--150 (2005).
- 11 Raine, N. E. & Chittka, L. The adaptive significance of sensory bias in a foraging context: Floral colour preferences in the bumblebee *Bombus terrestris*. *PLOS ONE* **2**, e556, doi:10.1371/journal.pone.0000556 (2007).
- 12 Gómez, J. M. *et al.* Spatial variation in selection on corolla shape in a generalist plant is promoted by the preference patterns of its local pollinators. *Proceedings of the Royal Society B: Biological Sciences* **275**, 2241-2249, doi:doi:10.1098/rspb.2008.0512 (2008).
- 13 Carazo, P. & Font, E. 'Communication breakdown': the evolution of signal unreliability and deception. *Anim. Behav.* **87**, 17-22, doi:<https://doi.org/10.1016/j.anbehav.2013.10.027> (2014).
- 14 Dawkins, M. S. & Guilford, T. The corruption of honest signalling. *Anim. Behav.* **41**, 865-873, doi:[https://doi.org/10.1016/S0003-3472\(05\)80353-7](https://doi.org/10.1016/S0003-3472(05)80353-7) (1991).
- 15 Alarcón, R., Davidowitz, G. & Bronstein, J. L. Nectar usage in a southern Arizona hawkmoth community. *Ecol. Entomol.* **33**, 503-509, doi:10.1111/j.1365-2311.2008.00996.x (2008).
- 16 Hansen, D. M., Olesen, J. M., Mione, T., Johnson, S. D. & Muller, C. B. Coloured nectar: distribution, ecology, and evolution of an enigmatic floral trait. *Biol. Rev. Camb. Philos. Soc.* **82**, 83-111, doi:10.1111/j.1469-185X.2006.00005.x (2007).
- 17 Johnstone, R. A. & Grafen, A. Dishonesty and the handicap principle. *Anim. Behav.* **46**, 759-764, doi:<https://doi.org/10.1006/anbe.1993.1253> (1993).
- 18 Schaefer, H., Schaefer, V. & Levey, D. How plant-animal interactions signal new insights in communication. *Trends Ecol. Evol.* **19**, 577-584, doi:10.1016/j.tree.2004.08.003 (2004).
- 19 Smith, M. J. & Harper, D. G. C. Animal signals: Models and terminology. *J. Theor. Biol.* **177**, 305-311, doi:<https://doi.org/10.1006/jtbi.1995.0248> (1995).
- 20 Guilford, T. & Dawkins, M. S. Receiver psychology and the evolution of animal signals. *Anim. Behav.* **42**, 1-14, doi:[https://doi.org/10.1016/S0003-3472\(05\)80600-1](https://doi.org/10.1016/S0003-3472(05)80600-1) (1991).
- 21 Essenberg, C. J. Intraspecific relationships between floral signals and rewards with implications for plant fitness. *AoB PLANTS* **13**, doi:10.1093/aobpla/plab006 (2021).
- 22 Riffell, J. A. *et al.* Behavioral consequences of innate preferences and olfactory learning in hawkmoth-flower interactions. *Proc Natl Acad Sci U S A* **105**, 3404-3409, doi:10.1073/pnas.0709811105 (2008).